# Signal neutrality, scalar property, and collapsing boundaries as consequences of a learned multi-timescale strategy

**Luca Manneschi**[1☉]*, **Guido Gigante**[2,3☉], **Eleni Vasilaki**[1,4], **Paolo Del Giudice**[2,3†]

**1** Department of Computer Science, University of Sheffield, Sheffield, United Kingdom, **2** Istituto Superiore di Sanità, Rome, Italy, **3** INFN, Sezione di Roma, Rome, Italy, **4** Institute of Neuroinformatics, University of Zurich and ETH Zurich, Switzerland

☉ These authors contributed equally to this work.
† Deceased
* l.manneschi@sheffield.ac.uk

**Data Availability Statement:** The source code and data used to produce the results and analyses presented in this manuscript are available at

## Abstract

We postulate that three fundamental elements underlie a decision making process: perception of time passing, information processing in multiple timescales and reward maximisation. We build a simple reinforcement learning agent upon these principles that we train on a random dot-like task. Our results, similar to the experimental data, demonstrate three emerging signatures. (1) signal neutrality: insensitivity to the signal coherence in the interval preceding the decision. (2) Scalar property: the mean of the response times varies widely for different signal coherences, yet the shape of the distributions stays almost unchanged. (3) Collapsing boundaries: the "effective" decision-making boundary changes over time in a manner reminiscent of the theoretical optimal. Removing the perception of time or the multiple timescales from the model does not preserve the distinguishing signatures. Our results suggest an alternative explanation for signal neutrality. We propose that it is not part of motor planning. It is part of the decision-making process and emerges from information processing on multiple timescales.

## Author summary

Humans and animals integrate sensory information before making a decision. The integration rate varies depending on the task. While driving could require quick reactions, evaluating the authenticity of a painting typically requires long observations. Consequently, the concept of representations created over multiple timescales appears necessary. Nevertheless, there is a lack of theoretical research that exploits multiple timescales, despite experimental evidence for the variety of integration rates. We, therefore, developed a decision-making model based on simple integrators with multiple characteristic times. We analysed its behaviour on a highly volatile, biologically relevant task. Through reward maximisation based on trial and error, the model discovers an effective strategy that is surprisingly different and more robust than the "classical" single timescale approach. This learned strategy exhibits a remarkable agreement with experimental

https://github.com/GuidoGigante/Signal-neutrality-scalar-property-and-collapsing-boundaries---Code.

**Funding:** EV acknowledges the support from a Google Deepmind Award. EV was funded by the Engineering and Physical Sciences Research Council (Grant Nos. EP/V055720/1, EP/V006339/1, EP/S030964/1, and EP/P006094/1). LM acknowledges the support from the Engineering and Physical Sciences Research Council (Grant No. EP/V006339/1). PD and GG were partially funded by the European Union Horizon 2020 Research and Innovation program under the FET Flagship Human Brain Project (SGA2 Grant agreement No. 785907 and SGA3 Grant agreement No. 945539). The funders had no role in study design, data collection and analysis, decision to publish, or preparation of the manuscript.

**Competing interests:** The authors have declared that no competing interests exist.

findings, suggesting a fundamental role of multiple timescales for decision-making. Our abstract model achieves a degree of biological realism while performing robustly in different environments.

# 1 Introduction

Perceptual decision-making is one of the most fundamental interactions of a biological agent with its environment. Perceptual decision-making processes have been long studied in the context of operant conditioning [1]. In these scenarios, an animal learns to associate choices and consequences by trial and error. Sub-optimal performance is considered a consequence of imperfect learning or the reflex of the learning strategy itself [2].

Outside this context, the research on perceptual decision-making has mainly focused on tasks where uncertainty (typically in the form of noisy signals) and time (e.g., duration of the observation and response delays) play a pivotal role [3–7]. In such scenarios, the errors made by the subject at the end of a training phase, as well as the relevant performance metrics (*e.g.* accuracy or speed of response), are deemed informative of the cognitive mechanisms involved [8–11]. There have been numerous attempts to compare the behaviour of animal subjects to the performance of different algorithms and determine how optimal the displayed behaviour is [8, 12–16].

One of the key ideas in perceptual decision-making is accumulating evidence over time [6, 8, 17–20]. The drift-diffusion model (also known as the 'bounded evidence accumulation' model) consists of two or more competing traces. These traces accumulate sensory evidence for different choices; the first trace to hit a threshold makes the associated option the final decision [21]. The drift-diffusion model is a continuous-time variant of the sequential probability ratio test [22, 23]. In the case of two-alternative forced choices, it is optimal in selecting between two hypotheses. Despite its simplicity, this model accounts for many psychophysical and neural observations. Examples are the distribution of response times and performance when varying sensory coherence [22, 23].

Notwithstanding its success, there are several alternatives to the standard drift-diffusion model [8, 24, 25] to account for unexplained phenomena such as primacy and recency effects, asymptotic accuracy, and "fast errors" [26–28]. Of notable importance is the Ornstein–Uhlenbeck model, which modifies the standard drift-diffusion model by including a decay term in the dynamics of the accumulation. Although the Ornstein–Uhlenbeck model can account for many experimental observations, including neurophysiological ones [24, 28], it introduces a characteristic timescale over which the model 'forgets' the past sensory information. A common approach in the literature is to treat the timescale of the accumulation as a free parameter that is optimised to match experimental data [28, 29].

Here we take a different approach. We study a decision-making problem within the context of reinforcement learning. The task is is intended to mimic a typical perceptual decision making setup [30]: an actor-critic agent has the task of determining whether a noisy signal has a positive or negative mean value. This agent can also decide when to decide, i.e., it can choose to wait instead of making a decision. We, thereby, postulate that the concept of reward maximisation is inherent in such problems.

Whilst not theoretically impossible, it is not straightforward to devise a biologically plausible mechanism to tune a single timescale parameter to the statistics of a task. To circumvent this issue, we propose a more biologically plausible process. The agent receives the signal from multiple integrators, each with a different time constant. Via reinforcement learning, the agent

learns how to weigh them appropriately to maximise the collected reward. We hypothesise that multiple timescales lead to robust performance across different tasks since it is unrealistic to expect one time constant to fit any problem. In the context of our model, we will explore robustness when varying the task difficulty, i.e. the signal to noise ratio, and contrast it with models of one time constant.

Beyond the computational advantage, such approach is consistent with the ample evidence of the coexistence of many timescales in brain functionality [31–35], even at the single neuron level [36–38], and for reward memory in reinforcement learning [39].

Another fundamental element of our model is that the agent perceives the passage of time. The agent has a "clock" available, several integrators with various time constants that increase by a fixed amount at each time step. In our model, we pair the clock's time constants with the time constants of the signal integrators. We do this to facilitate our mathematical analysis. However, we expect multiple time constants in the clock to implement a scalable population code for time, akin to what experimentally observed [40]. And, more specifically, to allow for more complex decision-making boundaries. We contrast an agent without any clock mechanism, an agent with a "single time constant" clock, and an agent with a multiple timescales clock. Our results highlight the performance advantages that a multiple timescales clock brings in.

We evaluate our agent concerning three properties observed in experimental data or theoretical analyses of decision-making processes. (1) Signal neutrality. We use this term as a shorthand to denote the observation that, for several hundred of milliseconds before the decision, the neurons in the lateral intraparietal cortex that correlate with the decision show the same response to different signal-to-noise ratios, with a time course of the firing activity that is indistinguishable in the different cases [5, 41]. One prior explanation is that the signals in that stage prepare the motor action. Here we evaluate this behaviour as part of the decision making process. (2) Scalar property or Weber's law [42]. The coefficient of variation (CV, the ratio of the standard deviation to the mean) remains constant as the task difficulty varies. (3) Collapsing boundaries. In the beginning, the agent should wait to integrate information to make an informed decision. However, the decision time is not unlimited; as time passes, the decision boundaries decrease to force the agent to act.

Our setup has similarities to a Partially Observable Markov Decision Process [43] with opportunity costs. The agent cannot access the real state of the world (in our case, the mean of the noisy signal and the time elapsed from the beginning of the trial). Instead, it has access to several observations at each time step. These observations are continuous variables that integrate noisy information about the state in terms of signal information and the time passed. These observations progressively correlate with the world's true state as the integration filters out the noise. The option to defer this decision in case of insufficient evidence complements the desirable action to find the sign of the stimulus. Yet, the presence of a time limit effectively imposes a cost on deferring the decision to accumulate more evidence.

## 2 Methods

### 2.1 Task definition

Inspired by classical random dots experiments [30], we model a two-alternative forced-choice task as a decision over the sign of the mean value of a noisy signal s(t) (see Fig 1). The signal (black line) consists of independent samples from a Gaussian distribution of mean $\mu$ and standard deviation $\sigma$, each drawn every time step $\Delta t = 10$ ms.

The agent is not required to decide at a prescribed time, it has the option to wait and then see another sample, or to perform one of two actions, 'left' and 'right', respectively associated with the decision $\mu < 0$ and $\mu > 0$ at each step. When an action is made, the episode ends, and

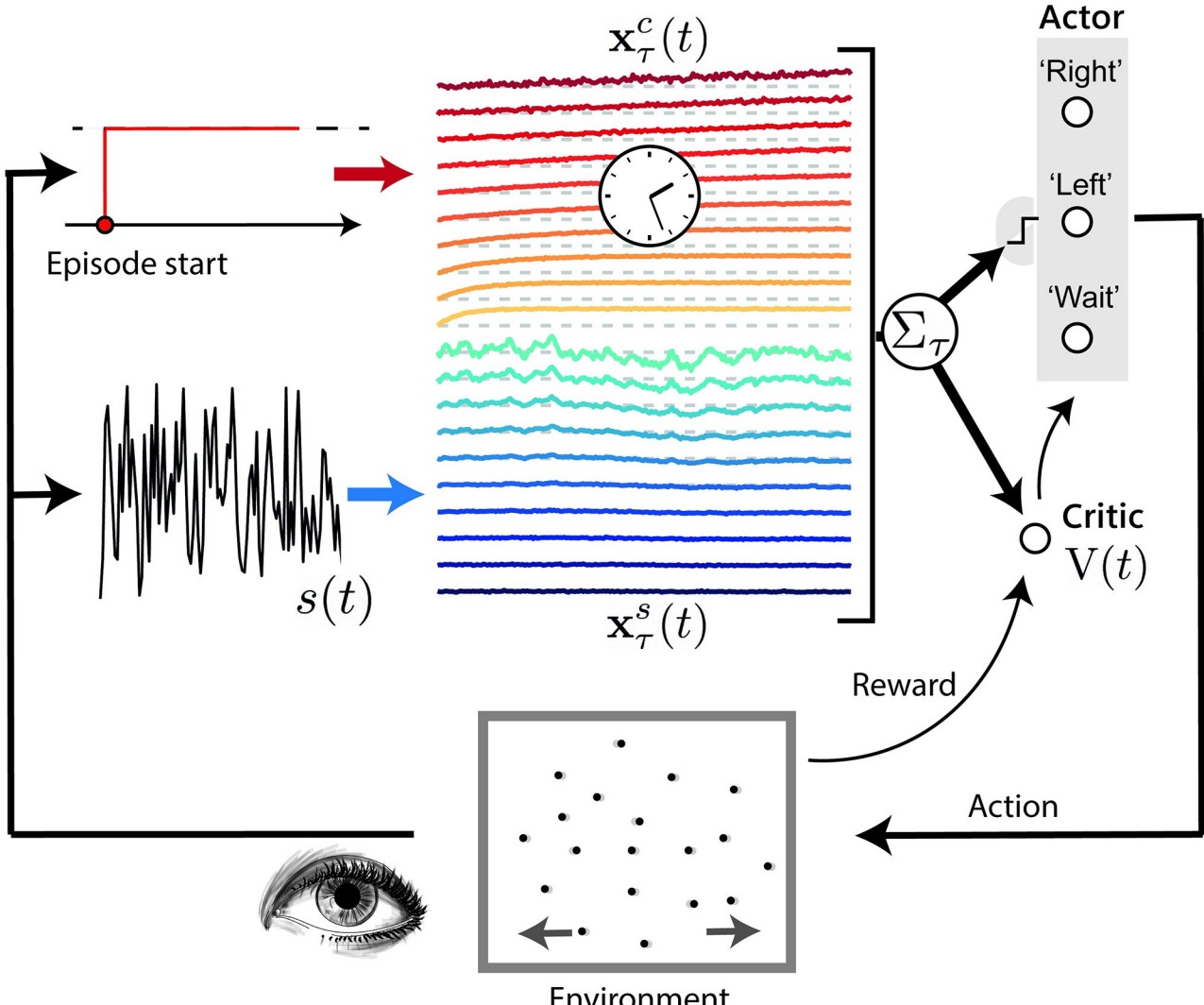

**Fig 1. Task and model schematic.** The environment corresponds to the random movement of a group of dots on a screen, which is represented as a uni-dimensional noisy signal $s(t)$ (black line), sampled at discrete time steps $\Delta t = 10$ ms from a Gaussian distribution of mean $\mu$ and variance $\sigma^2$. The task requires the subject to guess the sign of $\mu$, by moving a lever to the right (positive sign) or to the left (negative sign); the subject can 'choose when to choose', within a maximum episode duration $T_{\max}$. The learning agent integrates the signal over different timescales $\tau$ ($x_\tau^s(t)$s, blue lines); the agent integrates a constant input (depicted in red as a constant from the start of the episode) over the same timescales ($x_\tau^c(t)$s, yellow-red lines) to simulate an internal clock mechanism estimating the passage of time. In both cases, the darker the colour the longer the corresponding timescale. At each time instance, the weighted sums of the integrators (far right) are fed into a decision layer (the actor) that computes the probability of choosing 'left' and 'right', thus terminating the episode, or to 'wait' to see another sample of $s(t)$. If the subject gives the correct answer (the guessed sign coincides with the actual sign of $\mu$) within the time limit, a reward is delivered; otherwise, nothing happens. In any case, a new episode starts. The agent learns by observing the consequences (obtained rewards) of its actions, adapting the weights assigned to the $x_\tau^s(t)$s and $x_\tau^c(t)$s. During learning, the model estimates at each step $t$ the total future expected reward $V(t)$ (the critic) for the current episode as a linear summation of the integrators. Learning of the parameters is accomplished through a standard actor-critic reinforcement learning model, where the reward delivered by the environment is used to update the $V$ value function, which is then used to update the actor's parameters (see S1 Text for more details)

a reward is delivered only if the agent correctly guessed the sign of $\mu$; otherwise, the agent receives nothing. Each episode has a maximum duration $T_{\max}$. When $T_{\max}$ is reached, another 'wait' from the agent leads to the end of the episode and no reward is delivered.

Whilst $\sigma$ is constant, the value of $\mu$ is instead re-sampled at the beginning of each episode from a Gaussian distribution $p(\mu)$ of zero mean and variance $\sigma_\mu$. This second-order

uncertainty makes the agent experience a wide range of values of $\mu$, putting severely to the test its ability to generalise to episodes of varying signal-to-noise ratios.

## 2.2 Relationship between $\mu$ and random dots coherence

In random dots experiments, usually a number of dots moves randomly on a screen, with a fraction of them moving instead coherently in one direction (either left or right in different episodes). The percentage of coherently moving dots ('coherence') is a measure of how difficult an episode is, not unlike $|\mu|$ in the model (with sign of $\mu$ corresponding to a coherent movement towards left or towards right respectively). To make the parallel between the present task and the experimental settings more evident, in the following we will show results using either $|\mu|$ or the coherence of the signal, the two measures being related by:

$$|\mu| = 0.216 \frac{\text{coherence}}{\sqrt{100 - \text{coherence}}}. \tag{1}$$

In fact, in [5], every three frames on the screen, a fraction $c$ ('coherence') of dots are moved coherently in the chosen direction by $dx$, while the other $1 - c$ dots are randomly displaced. We assume that each of the randomly moving dots is subjected to a change $\Delta x$ in their position following a probability distribution, with $\langle dx \rangle = 0$ and $\text{Var}[dx] = \sigma_x^2$. Imagining that neurons with different receptive fields help to estimate the average movement of the dots at each time step, we end up with a signal $s$ of mean:

$$\mu \equiv \langle s \rangle = c \, dx \tag{2}$$

and variance:

$$\sigma^2 \equiv \text{Var}[s] = (1 - c) \, \sigma_x^2 \tag{3}$$

Then, we have the relationship:

$$\frac{\mu}{\sigma} = \frac{c}{\sqrt{1 - c}} \frac{dx}{\sigma_x} \tag{4}$$

or:

$$\mu \propto \frac{\text{coherence}}{\sqrt{100 - \text{coherence}}}, \tag{5}$$

where we have expressed the coherence as a percentage. Eq 1 is a special case of this one, with a proportionality constant chosen to match experimental ranges.

## 2.3 An agent over multiple timescales

The section is dedicated to the definition of the proposed model. In contrast to previous research works on the decision making process, the agent makes decisions thanks to a reservoir of multiple timescales of integration and an estimate of the passage of time. The agent comprises $n_\tau = 10$ leaky integrators $x_\tau^s$ (dark blue to cyan lines in Fig 1) that independently integrate the noisy signal $s(t)$ over different timescales $\tau$:

$$\dot{x}_\tau^s = -\frac{x_\tau^s - s(t)}{\tau}, \tag{6}$$

and correspondingly $n_\tau$ leaky integrators $x_\tau^c$ (yellow to red lines in Fig 1) that integrate a constant input (a 'time signal', here valued 1), to account for the possible effects of an internal

'clock':

$$\dot{x}_\tau^c = -\frac{x_\tau^c - 1}{\tau}. \tag{7}$$

Both the $x_\tau^s$ and the $x_\tau^c$ are reset to 0 at the beginning of each episode (note, therefore, that $x_\tau^c(t) = 1 - e^{-\frac{t}{\tau}} \geq 0$ for all $t$). Moreover, we added noise to the values of the integrators at a given time (Eqs 6 and 7) redefining:

$$x_\tau^s(t) \leftarrow x_\tau^s(t) + \xi_\tau^s \tag{8}$$

$$x_\tau^c(t) \leftarrow x_\tau^c(t) + \xi_\tau^c \tag{9}$$

$\xi_\tau^s(t)$ and $\xi_\tau^c(t)$ are drawn independently for each $t$ and each $\tau$ from a Gaussian distribution with zero mean and standard deviation $\sigma_I$. The $\xi_\tau^s(t)$s and $\xi_\tau^c(t)$s are introduced to model the intrinsic noise implied in any plausible biological implementation of the integration process, such as fluctuations in the instantaneous firing rate of a network of neurons.

The $\tau$s are chosen on a logarithmic scale (*i.e.*, $\tau_i = \alpha\,\tau_{i-1}$, with $\alpha$ a suitable constant), with $\tau_1 = \tau_{\min} = 100$ ms and $\tau_{n_\tau} = \tau_{\max} = 10$ s, so as to allow the agent to accumulate information over a wide range of different timescales. The specific choice of the distribution of timescales is not critical to the following results, assuming that the values of $\tau$s are densely spread over a wide range (see Results and S1 Text).

At each time step $t$, the agent computes six weighted sums, three for the signal $x_\tau^s(t)$ and three for the clock $x_\tau^c(t)$. The first four of these weighted sums are related to the two possible actions:

$$\Sigma_{\text{right}}^s(t) \equiv \sum_\tau \theta_{\text{right},\tau}^s\, x_\tau^s(t) \tag{10}$$

$$\Sigma_{\text{right}}^c(t) \equiv \sum_\tau \theta_{\text{right},\tau}^c\, x_\tau^c(t) + b_{\text{right}} \tag{11}$$

$$\Sigma_{\text{left}}^s(t) \equiv \sum_\tau \theta_{\text{left},\tau}^s\, x_\tau(t) \tag{12}$$

$$\Sigma_{\text{left}}^c(t) \equiv \sum_\tau \theta_{\text{left},\tau}^c\, x_\tau^c(t) + b_{\text{left}} \tag{13}$$

where $b_{\text{right}}$ and $b_{\text{left}}$ are constants and can be described as the propensity of the agent to make the corresponding actions before the beginning of an episode. The $\Sigma^s$s and the $\Sigma^c$ carry information, respectively, on the signal and the time elapsed since the beginning of each episode. Even though the $x_\tau^c$ increase with time, the $\Sigma^c$s can be non-monotonic, something that will play an important role in implementing an effective 'moving threshold' for the decision mechanism.

The other two sums are instead related to the 'wait' option:

$$\Sigma_{\text{wait}}^s(t) \equiv \sum_\tau \theta_{\text{wait},\tau}^s\, |x_\tau^s(t)| \tag{14}$$

$$\Sigma_{\text{wait}}^c(t) \equiv \sum_\tau \theta_{\text{wait},\tau}^c\, x_\tau^c(t) + b_{\text{wait}}, \tag{15}$$

where the absolute value in Eq 14 is taken to account for the intuition that a signal and its negative mirror should equally affect the agent's propensity to defer a decision. The constant $b_{\text{wait}}$ has similar meaning to the biases $b_{\text{right}}$ and $b_{\text{left}}$, but related to the 'wait' action. By setting:

$$\Sigma_x \equiv \Sigma_x^s + \Sigma_x^c \tag{16}$$

(with $x \in \{\text{left, right, wait}\}$), the six sums are then non-linearly combined through a softmax function (the circles corresponding to the actor on the right of Fig 1) to define a probability distribution over the possible actions:

$$p_{\text{right}}(t) = \frac{e^{\Sigma_{\text{right}}(t)}}{e^{\Sigma_{\text{left}}(t)} + e^{\Sigma_{\text{wait}}(t)} + e^{\Sigma_{\text{right}}(t)}} \tag{17}$$

and analogous expressions for 'left' and 'wait'. By definition, $p_{\text{left}}(t) + p_{\text{wait}}(t) + p_{\text{right}}(t) = 1$ for every $t$. The agent then randomly chooses an option according to the three probabilities.

The agent is thus completely determined by the choice of the six sets of $n_\tau$ weights: $\theta_{\text{left},\tau}^s$, $\theta_{\text{wait},\tau}^s$, $\theta_{\text{right},\tau}^s$, $\theta_{\text{left},\tau}^c$, $\theta_{\text{wait},\tau}^c$, $\theta_{\text{right},\tau}^c$, and three constant offsets $b_{\text{left}}$, $b_{\text{wait}}$, and $b_{\text{right}}$. We note how this set of parameters is redundant, because of the way they enter Eq 17. For example, we could make the substitution $b_{\text{right}} \leftarrow b_{\text{right}} - b_{\text{wait}}$, $b_{\text{left}} \leftarrow b_{\text{left}} - b_{\text{wait}}$, and $b_{\text{wait}} = 0$ and the resulting agent would be mathematically equivalent to the original one. We use such redundant definition in order to simplify the description of the model, making it the most symmetric for 'left', 'right', and 'wait'. These weights and offsets are learned by trial-and-error through a reinforcement learning procedure aiming to maximise reward. All the results shown, if not otherwise stated, are obtained using the same set of weights, at the end of the training procedure, with $T_{\max} = 2$ s, $\sigma = 0.18 \, \text{s}^{-\frac{1}{2}}$, $\sigma_\mu = 0.25$, and $\sigma_I = 0.02$. Training of the parameters of the model is achieved through a standard actor-critic reinforcement learning algorithm [43], which is described in S1 Text. During learning, the model estimates at each step $t$ the total future expected reward V($t$) for the current episode. Such estimate is computed by a linear summation of the integrators (Fig 1, bottom-right) and is used to establish a moving baseline to modulate the changes in the model's weights during training. The parameters of the actor and the critic are then updated thanks to the utilisation of eligibility traces [43].

## 2.4 Comparative models

To understand the role of multiple timescales and of the internal clock in the results, we compare the performance of the proposed agent with other decision making models.

1. *Single integrator with optimised threshold.* This refers to the Ornstein-Uhlenback decision process [24, 28], which is a generalisation of the standard drift diffusion model [21]. The model is composed by an integrator with one timescale and a threshold. The dynamic of the integrator is given by Eqs 6 and 8. A decision is triggered when the latter activity reaches ± a threshold value $\Theta_\tau$. In our case, the action 'right' is made when $x_\tau^s(t) \geq \Theta_\tau$, while the agent performs the 'left' action when $x_\tau^s(t) \leq -\Theta_\tau$. Considering the presence of a single timescale of integration, we will consider multiple versions of the process, each with a different value of $\tau$. For each model with a specific $\tau$, the threshold $\Theta_\tau$ will be optimised through grid search by maximising the accuracy on the considered task. In this way, we are certain that the process will exhibit the highest possible performance on the considered task, or performance that are negligibly distant to its theoretical optimal.

2. *Agent with a single timescale.* The model refers to a reinforcement learning agent similar to the proposed one, but with only one timescale of integration. Practically, the agent definition is again based on Eqs 10–14, but every summation over $\tau$ reduces to a single term. The

total number of parameters in this case is thus nine ($\theta^s_{\text{left}}$, $\theta^s_{\text{wait}}$, $\theta^s_{\text{right}}$, $\theta^c_{\text{left}}$, $\theta^c_{\text{wait}}$, $\theta^c_{\text{right}}$, $b_{\text{left}}$, $b_{\text{wait}}$, and $b_{\text{right}}$). As for the single integrator with optimised threshold, we will simulate multiple versions of the model to vary the timescale of integration $\tau$. We note how, in contrast to the previous comparative model, this process has an estimate of the passage of time over one single time constant. For this feature, the process departs from the other decision making models in the literature. This agent will help us to understand the role of multiple timescales further, providing a baseline where a basic knowledge of the internal clock is present, but where integration occurs over a single $\tau$.

3. Agent with multiple signal integrators, but without internal clock. The model is again defined by Eqs [10]–[14], but without temporal information, that is $\theta^c_{\text{left}} = \theta^c_{\text{wait}} = \theta^c_{\text{right}} \equiv 0$. The model will constitute an additional comparison to separate the roles of the availability of multiple timescales on the signal and on the internal clock mechanism.

Because of the presence of multiple integrators, the proposed agent effectively lowers the total noise by summing up $n_\tau$ integrators $x^s_\tau$ affected by independent sources of noise $\xi^s_\tau$ ([Eq 8]). Thus, when comparing the proposed agent with one of the above models that exploits a single time constant, we rescaled the amount of noise $\sigma_I$ affecting the single integrator by a factor $\alpha_I$, defined as

$$\alpha_I = \frac{1}{\sqrt{\sum_\tau \theta^2_{*,\tau} / \max_\tau(\theta^2_{*,\tau})}} \leq 1 \tag{18}$$

where $\theta_{*,\tau}$ refers to the optimal weights $\theta_{\text{right},\tau}$ found after training of the proposed model (we could equivalently use the optimal $\theta_{\text{left},\tau}$, since after training $\theta_{\text{left},\tau} \simeq -\theta_{\text{right},\tau}$, as it should be considering the symmetry of the task considered). Thus, $\alpha_I = 1$ when just one of the $\theta_{*,\tau}$ is different from 0, i.e. when the agent utilises just one integrator. On the other hand, the maximum $\alpha_I = \frac{1}{\sqrt{n_\tau}}$ is attained when the agent weights equally all the integrators. In this way, the total amount of noise in the single timescale model is effectively equivalent to the one present in the multiple timescales agent.

## 2.5 signal neutrality and scalar property measures

To measure signal neutrality, we take the average $\Delta\Sigma_{\text{right}}(t)$ (see [Eq 21]), aligned to decision time, for six different coherences (0%, 3.2%, 6.4%, 12.8%, 25.6%, 51.2%); each curve is considered for an interval between 0 and 600 ms before the decision is taken; if the number of points to average for a given coherence drops below 100 before the 600 ms, the interval of definition of that curve is shrunk accordingly. We then rescale all the curves to fit inside the range 0–1, so that the minimum of the minimum values attained by each curve is 0; and the maximum of the maxima is 1. Then we compute, for each time, the maximum distance between any pairs of rescaled curves (this distance is of course always $\leq 1$ thanks to the rescaling). Finally we take the average of such maximum distance, and take the inverse: this is the operative measure of signal neutrality used throughout the paper.

To give a measure of scalar property, we compute the coefficient of variation CV for the distribution of response times corresponding to six values of coherence (0%, 3.2%, 6.4%, 12.8%, 25.6%, 51.2%). We then take the inverse of the difference between the maximum and the minimum value of CV: this is the reported measure of the scalar property.

## 3 Results

First, we analyse how the behaviour of the optimised agent is different from the standard drift-diffusion model by exploiting integration over a variety of timescales. Fig 2 shows the evolution of $p_{\text{right}}(t)$ (blue line) and $p_{\text{left}}(t)$ (red) during an episode where the correct action is 'right' (that is, $\mu > 0$). As expected, $p_{\text{right}}(t)$ is for the most part greater than $p_{\text{left}}(t)$ (although this is unnoticeable in the plot where the probabilities are very small), signalling that the agent favours the action associated with the correct decision. Nevertheless, both probabilities are very low most of the time, implying that $p_{\text{wait}}(t)$ is often close to one (not shown). Thus, the agent appears to select a strategy in which decisions are made within short 'active' windows of time during which fleeting bursts of $p_{\text{left}}(t)$ or $p_{\text{right}}(t)$ make an action possible. Such strategy is not trivially associated with the intuitive picture of a process accumulating information over time until some threshold is met (for instance, see model 1. in section 2.4).

In fact, the agent exploits the information carried by the different integrators by waiting for their consensus, akin to a majority vote. A short-lived fluctuation in the fastest integrators would not be enough for a decision. Yet, in conjunction with a longer-lived fluctuation of the slower integrators, a burst in one of the actions is triggered. Being the consensus fleeting, such probability bursts are usually quite low (they often stay below a probability of 0.1) and

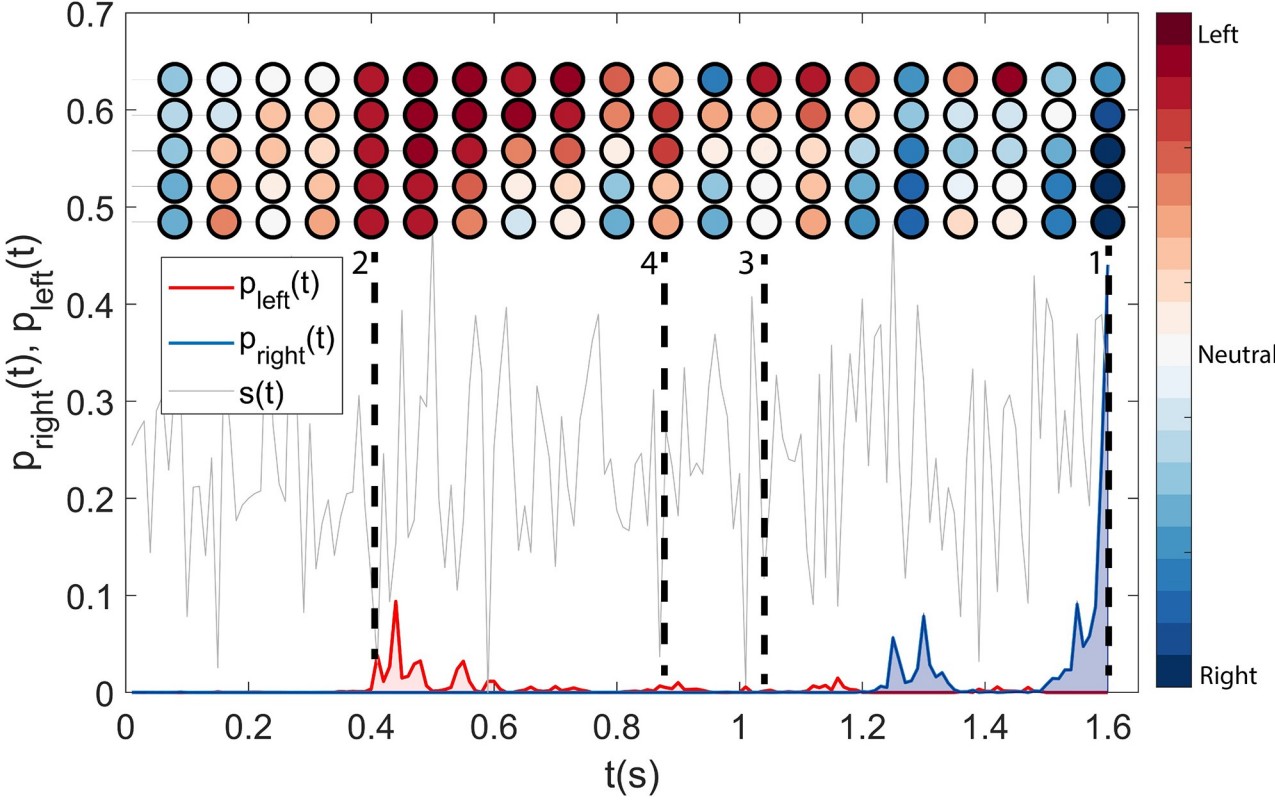

**Fig 2. Learned decision strategy.** Evolution of $p_{\text{right}}(t)$ (blue line) and $p_{\text{left}}(t)$ (red) during an episode (signal $s(t)$ in dashed grey) where the correct action is 'right' (that is, $\mu > 0$). Decisions are made within short 'active' windows of time during which fleeting bursts of $p_{\text{left}}(t)$ or $p_{\text{right}}(t)$, corresponding to the alignment of many integrators, make an action possible. The coloured circles correspond to the values of a subset of 5 of the 10 integrators (slow to fast associated timescales from top to bottom). The colours (blue to red) represent the 'tendency' of an integrator toward a decision. Blues correspond to positive (toward the 'right' action) values, while reds to negative values (toward the 'left' action). These tendencies are computed using the average behaviour of the specific integrator as a reference value. In other words, if the circle is blue, it means that the value assumed by the integrator was higher than usual at that specific time. Uniformly positive (negative) values for the integrators are associated with bursts of $p_{\text{right}}$ ($p_{\text{left}}$, see times denoted with 1 and 2 in the plot). Outside bursts (point 3) or when a burst withers (point 4), not all the integrators assume low absolute values.

therefore function as 'open windows' paving the way to a decision, more than as 'funnels' forcing it. Decisions therefore happen when the different timescales stay in agreement for an extended period (roughly 100 ms).

This is illustrated in Fig 2 with coloured circles, each row representing the evolution of one integrator (for a subset of 5 of the 10 integrators, with slow to fast timescales from top to bottom). As expected, inside a burst of $p_{\text{right}}(t)$ almost all the integrators present large positive values (dark blue, see for example temporal instance number 1 in Fig 2). On the other hand, integrators typically assume negative values (light to dark red) in correspondence of bursts of $p_{\text{left}}(t)$, as it is shown in the temporal instance number 2. The converse is not true: in absence of probability bursts, not all the integrators assume low absolute values (see, for example, coloured circles corresponding to number 3). This is due to the fact that the integrators, though correlated, detect fluctuations in the signal over different timescales. Moreover, the non-linear nature of the probability function (Eq 17) dampens integrators' fluctuations falling below a given range of values. When a burst fades away (see for example points between 2 and 4) not all the integrators go down together. Initially the faster integrators become neutral or even slightly change sign. Afterwards the slower integrators follow suit. Of course, the process is not completely linear, and intermediate integrators can assume (see instance number 4 and neighbouring points) higher values, while the slowest (fastest) ones are still decreasing (fluctuating rapidly). A more detailed analysis of the behaviour of the agent can be found in S1 Text and S1 Fig.

## 3.1 Model's performance

Fig 3A shows the fraction of correct choices as a function of the decision time, both for the agent at the end of training (black line) and for the optimal fixed-$t$ observer (blue line) that, at each time $t$, simply chooses according to the sign of the sum of the signal up to time $t$. The latter's performance can be derived analytically:

$$\text{Fraction Correct}(t) = \frac{1}{2} + \frac{1}{\pi} \arctan \sqrt{\frac{\sigma_\mu^2 t}{\sigma^2}} \tag{19}$$

If the task were to decide exactly at time $t$, no other decision maker could outperform it; for this reason it is deemed optimal. The comparison with the fixed-$t$ observer sheds light on the agent's strategy and the underlying trade-offs.

The agent is free to "choose when to choose", thus it is not surprising that its performance is higher than the optimal fixed-$t$ observer for shorter decision times (the inset of Fig 3A shows the distribution of decision times for the agent). We see that the two performances cross slightly above the average decision time for the agent. Beyond this point, the fixed-$t$ observer dominates. Indeed, the agent can make the easy decisions early on and wait to see how the signal evolves when the choice appears more uncertain. In contrast, the fixed-$t$ observer is bound to decide at time $t$, no matter how clear or ambiguous the observed signal was up to that point. The steep rise of the agent's performance for very short decision times is mainly a reflection of its ability to tell apart the easy episodes from the hard ones. The fixed-$t$ observer catches up for longer times, where the agent is left with only the most difficult decisions and its performance consequently declines. For the fixed-$t$ observer, instead, larger $t$s always mean more information and therefore its performance monotonically increases. We notice how at the crossing point, the agent has already made the large part of its decisions, as it is apparent from the distribution of decision times.

Fig 3B shows how the agent (horizontal line) outperforms all the single integrators with optimised thresholds (circles, see section 2.4 for the model definition). The performance of the

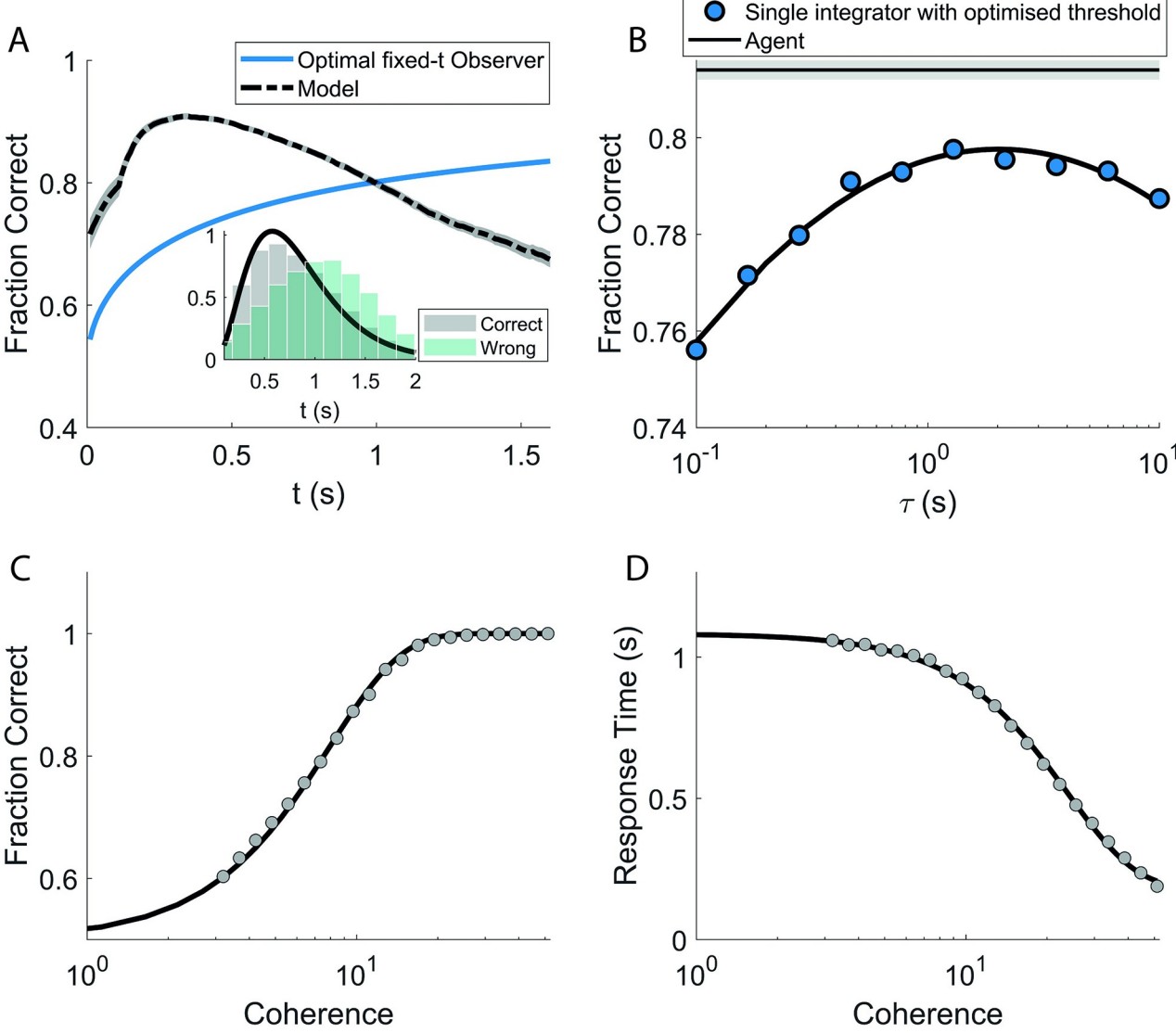

**Fig 3. Performance after training. A**: Fraction of correct choices as a function of the decision time, both for the agent at end of training (black line) and for optimal fixed-*t* observer (blue line) that simply chooses according to the sign of the accumulated signal up to time *t*(see text). The agent clearly outperforms the fixed-*t* observer for shorter decision times, thanks to its freedom to 'choose when to choose'. The steep rise of the agent's performance for very short decision times is mainly a reflection of its ability to tell apart the easy episodes from the hard ones. Inset: response time histograms for correct (grey) and wrong (green) decisions **B**: the agent (horizontal line) outperforms, considering the fraction of correct choices on a sample of episodes, all the single-timescale integrators with optimised decision threshold (dots; the continuous line is a second-degree polynomial fit for illustration purposes). The performance of the single-timescale integrator peaks for intermediate values of the associated timescale *τ*, though it always stays below the performance attained by the agent. The grey strip around the agent's line marks the 25%-75% of the values obtained for the performance upon 100 repetitions of the training procedure (see Fig 5 for further details). **C** and **D**: Accuracy and mean response times for different values of coherence (dots). **C**: The accuracy curve for the agent is in very good agreement with experimental findings: the black line is the result of a fit on experimental data ([5]; see text for more details). **D**: As accuracy increases, responses become faster, as found in experiments (black line: fit with a sigmoid-like function).

single-timescale integrator peaks for intermediate values of the associated timescale *τ*, though it always stays well below the performance attained by the agent. The agent, therefore, is able to leverage the information on multiple timescales from the signal and the internal clock to gain a clear performance advantage with respect to the drift-diffusion model on the whole

spectrum of $\tau$s. A more detailed comparison between the performance of the different models considered will be given in section 3.5.

Fig 3C and 3D show the accuracy and the mean response time of the agent as the coherence of the signal varies (Eq 1). The black line in panel Fig 3C is computed as:

$$\text{Fraction Correct(coherence)} = 1 - \frac{1}{2} \exp\left[-\left(\frac{\text{coherence}}{7.97}\right)^{1.62}\right] \qquad (20)$$

as in Fig 3 of [5], where the parameters of the curve were fitted to experimental data. The match between the experimental fit and the result of the agent is striking. In Fig 3D, instead, the black line is a generic sigmoidal function plotted for illustration purposes. As found in the experiments, the agent's responses become faster as the task becomes easier (larger coherences).

## 3.2 Signal neutrality

A more microscopic look at the decision process surprisingly uncovers shared features between the internal dynamics of the artificial agent and the activity observed in neurons in the lateral intraparietal cortex (LIP) during a random dots task [5, 41].

We now define a key observable of the model that will be central in the following (see Eqs 16, 10 and 11):

$$\Delta\Sigma_{\text{right}}(t) \equiv \Sigma_{\text{right}}(t) - \Sigma_{\text{wait}}(t) \qquad (21)$$

and its 'left' counterpart $\Delta\Sigma_{\text{left}}(t) \equiv \Sigma_{\text{left}}(t) - \Sigma_{\text{wait}}(t)$. Eq 21 ($\Delta\Sigma_{left}$) provides a direct measure of the propensity of the agent to make a 'right' ('left') decision at time $t$.

Fig 4A shows the evolution of $\Delta\Sigma_{\text{right}}$, averaged over many episodes in which the agent has made the correct decision 'right'. The traces are grouped by signal coherence. The left part of Fig 4A shows the evolution of the average $\Delta\Sigma_{\text{right}}$, with traces aligned to the beginning of the episode (onset of the external signal). $\Delta\Sigma_{\text{right}}$ shows a marked sensitivity to the coherence of the signal. Moreover, the traces do not saturate over several hundreds of milliseconds, highlighting how the agent is making use of its slower integrators.

Ramp-like changes in the discharge of LIP neurons have been repeatedly observed, with steeper rise in spike rate for higher stimulus coherence (see, *e.g.*, Figure 7 in [5]). Such ramps, originating in the extrastriate visual cortex in the case of LIP neurons, have been interpreted as a signature of the accumulation of evidence for or against a specific behavioural response [10, 17]. This interpretation is fully compatible with what is seen in the agent.

However, when the averages of the $\Delta\Sigma_{\text{right}}$ traces (or of the activity of LIP neurons) are performed by aligning the episodes to the time of the decision, a clear signature of signal neutrality emerges. The sensitivity to the stimulus's coherence is lost and all the lines surprisingly collapse on the same curve for several hundreds of milliseconds (Fig 4A, right). We emphasise that such collapse over an extended period of time is key to recognise signal neutrality: any decision model with a deterministic threshold, for example, would display a collapse at decision time (exactly at the threshold), but not necessarily at previous times; in this case, according to our definition, the model would not display signal neutrality.

For the experimental data, a reasonable explanation for such collapse is that the neuronal circuitry is engaged in stereotyped dynamics, independent from the signal, just after a decision is made and before it is manifested with a physical action, perhaps as the result of a feedback from downstream areas.

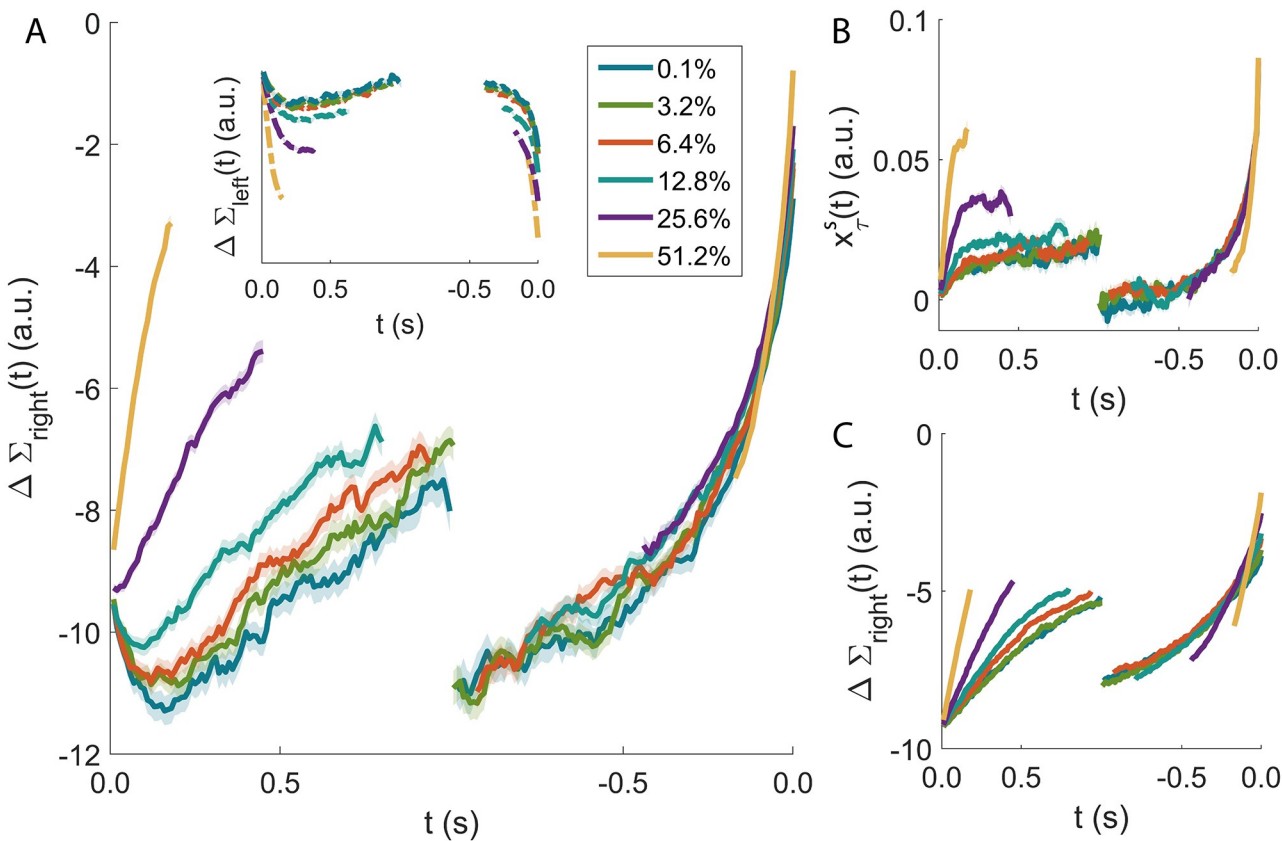

**Fig 4. signal neutrality.** $\Delta\Sigma_{\text{right}}(t)$ (see Eq 21) provides a direct measure of the propensity of the agent to make a 'right' decision at time $t$. **A** Evolution of $\Delta\Sigma_{\text{right}}$, averaged over many successful episodes with the same signal coherence. On the left, the episodes are aligned to the beginning of the episode and $\Delta\Sigma_{\text{right}}$ shows a marked sensitivity to the coherence of the signal. When the average is performed by aligning all the episodes to the time of the decision (right), signal neutrality clearly appears: the sensitivity to the signal strength is completely lost and all the lines collapse on the same curve for several hundreds of milliseconds. Inset: the same analysis on wrong episodes. The similarities with what is found in the discharge of LIP neurons during a motion-discrimination task are striking (see, *e.g.*, Figure 7 in [5]). **B**: Time course of $x_\tau^s$ for a single-timescale integrator with $\tau = 2s$ and optimised decision threshold ($x_\tau^s$, for an integrator with threshold, plays the role that $\Delta\Sigma_{\text{right}}$ has in the agent). **C**: Time course of $\Delta\Sigma_{\text{right}}$ (see Eq 21 for an agent optimised with a single timescale $\tau = 2s$). In both **B** and **C** the collapse of the curves for different signal coherences is imperfect (rightmost part of the plots).

But this cannot hold for the agent, where instead signal neutrality arises precisely from the presence of multiple timescales. Fig 4B and 4C show the time course of the equivalent of $\Delta\Sigma_{\text{right}}$ for the models with a single timescale (see Section 2.4). For both these models, we display the results obtained from an example time constants of $\tau = 2.0$ s. In the single integrator with optimised threshold, $x_\tau^s$ plays the role that $\Delta\Sigma_{\text{right}}$ has in the agent.

In the latter, the collapse of the curves for different signal coherences is not as evident (Fig 4B and 4C, rightmost part). To make this statement more systematic, we introduce an operative measure of signal neutrality. We computed the inverse of the maximum distance between the curves for different coherences averaged over an interval of up to 600 ms prior to the decision (see Methods). In Fig 5A we report this measure for the agent (horizontal line) and the models with a single timescale (coloured upper bars). The comparative models report lower values in terms of signal neutrality and accuracy (Fig 5B).

The propensity of the agent $\Delta\Sigma_{\text{left}}$ to make the erroneous 'left' decision does not display signal neutrality. The same holds true for its experimental counterpart, that is the activity of LIP

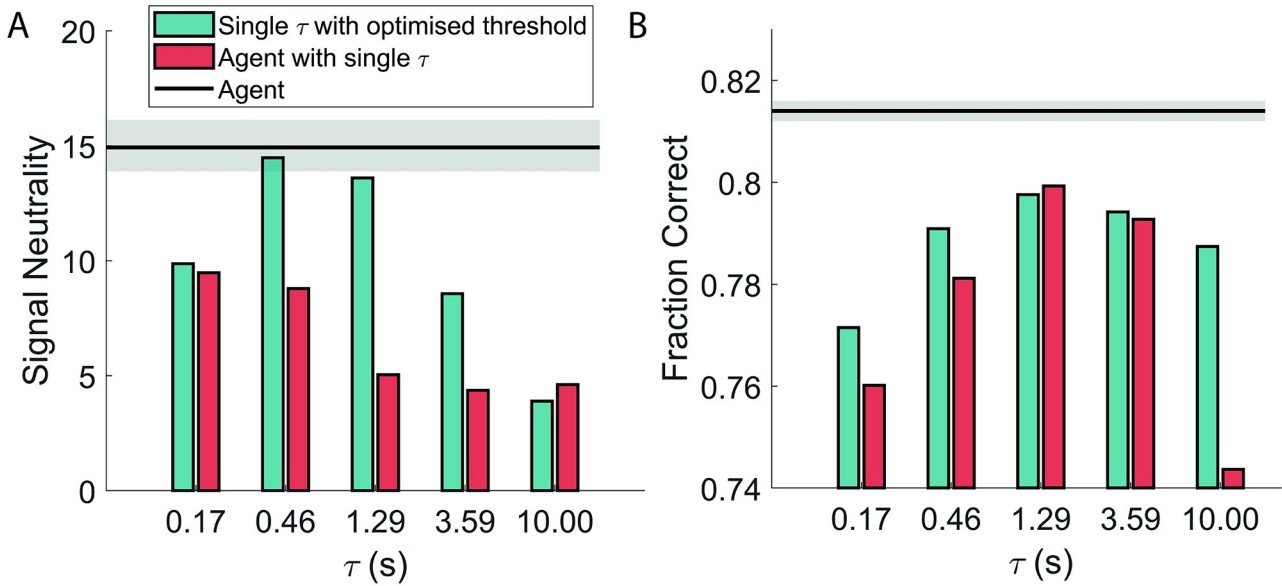

**Fig 5. Comparison of signal neutrality (A) and performance (B) for the single-τ agent and the single-timescale integrator as τ varies.** The proposed model (black horizontal lines) shows better accuracy while exhibiting the experimentally observed collapse of the time course of neuronal activity aligned at the decision time. The grey area marks the 25%-75% of the values obtained for each of the two observables upon 100 repetitions of the training procedure; more specifically, each independent training has been halted where signal neutrality peaked, conditioned to having already reached a performance of 0.81 or above; this translates to an average training length of about 73000 episodes (10%-90% range: 39000–110000).

neurons when the random dot motion is away from their receptive field (see Figure 7 in [5], dashed lines). Finally, the comparison between the models in Fig 4 emphasises how in our simulations the signal neutrality is a consequence of the availability of multiple timescales.

### 3.3 The scalar property

The agent's behaviour conforms to one of the hallmarks of temporal cognition: the scalar property or Weber's law for interval timing [42]. This is illustrated in Fig 6A, where the distributions of response times of the agent are shown for three different values of coherence. As the coherence increases, the average response time of the agent decreases from 4.6 s to 370 ms.

Simply stated, the scalar property—as observed for example in interval timing [42], and multistable perception [44]—implies that higher moments of the intervals' distribution scale as appropriate powers of the mean. This implies a constant coefficient of variation. In other words, the shape of the distribution does not change when its mean varies even over wide ranges.

Notwithstanding a mean value that varies by more than one order of magnitude, the coefficient of variation of the agent moves in a very narrow range which is compatible with the experimental findings [42, 44]. The invariance of the shape of the distribution is made immediately evident in the inset of Fig 6A. Here the fitted Gamma distributions (black lines in the main plot) are rescaled to have mean equal to 1. The similarity of the three curves is striking. Fig 6B shows the coefficient of variation CV as the coherence varies for the proposed agent (black) and the comparative models (blue and red colours, see Section 2.4 for more details). The coefficient of variation has an approximately constant value for the proposed agent only. We remark that an agent with a single integrator has information regarding the passage of time over a single time constant, and that the model depicted in blue has multiple integrators

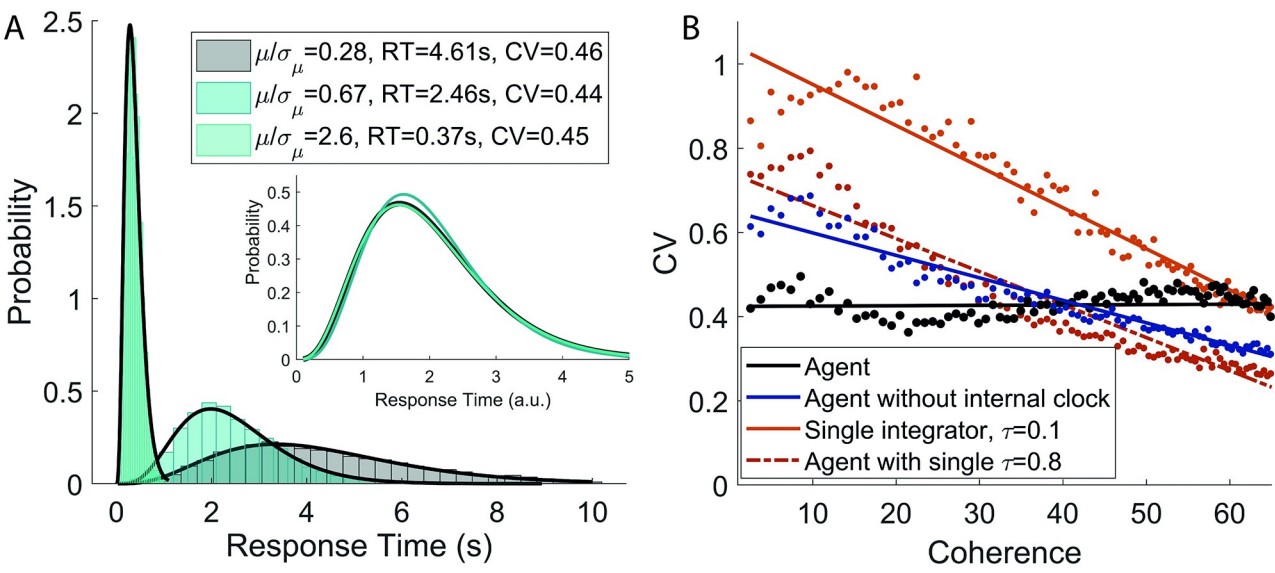

**Fig 6. Scalar property. A**: The average response time of the agent decreases as the signal coherence increases; still the coefficient of variation of the response times varies in a very narrow range (see legend). The black lines are the best fit of the simulation histograms with a Gamma distribution. Inset: the fitted Gamma distributions are rescaled to have mean equal to 1, making immediately evident how the shape of the distribution stays almost unchanged as its average moves over almost one order of magnitude (colours consistent with the histograms in the main plot). Note how the highest value of coherence is very unlikely under the distribution used for training the agent (corresponding to a value of $\mu$ five times the standard deviation $\sigma_\mu$ of the distribution of $\mu$). The 'invariant shape' property of the response time distribution therefore holds well beyond the typical range of functioning of the agent. **B**: Coefficient of variation (CV) of the different models as the coherence increases. The scalar property is satisfied exclusively by the proposed agent (black line). The single timescale models are reported with two different values of $\tau$. Other choices of $\tau$ give comparable results.

but lacks any explicit temporal information. Thus, the key ingredient for the scalar property is again the availability of multiple timescales, in particular on the estimate of the passage of time.

On the other hand, it is not surprising that the single integrator with optimised threshold is unable to display the scalar property. In fact, for the pure drift-diffusion model ($\tau = \infty$), the coefficient of variation as a function of the coherence $c$ can be computed analytically [45] (see also Eq 1):

$$ \mathrm{CV} = \left( \frac{100 - c}{c^2} \right)^{1/4}, \tag{22} $$

and it is clearly not constant.

Lastly, we note how the highest values of coherence reported in the plots are very unlikely under the distribution used during the training phase. A coherence of 50% roughly corresponds to a value of $\mu$ that is five times the standard deviation $\sigma_\mu$ of the distribution of $\mu$. Thus, the scalar property appears to be a very robust property of the learned decision strategy of the proposed agent, holding well beyond the range of functioning to which the agent has been accustomed during training.

In view of the above considerations, signal neutrality and the scalar property share a similar origin. Further evidence of this can be found in the evolution of the two measures during the training phase.

Fig 7 shows the average evolution of signal neutrality (black line; the same measure reported in Fig 4D), scalar property (blue line; see Methods for the definition of the metric), and

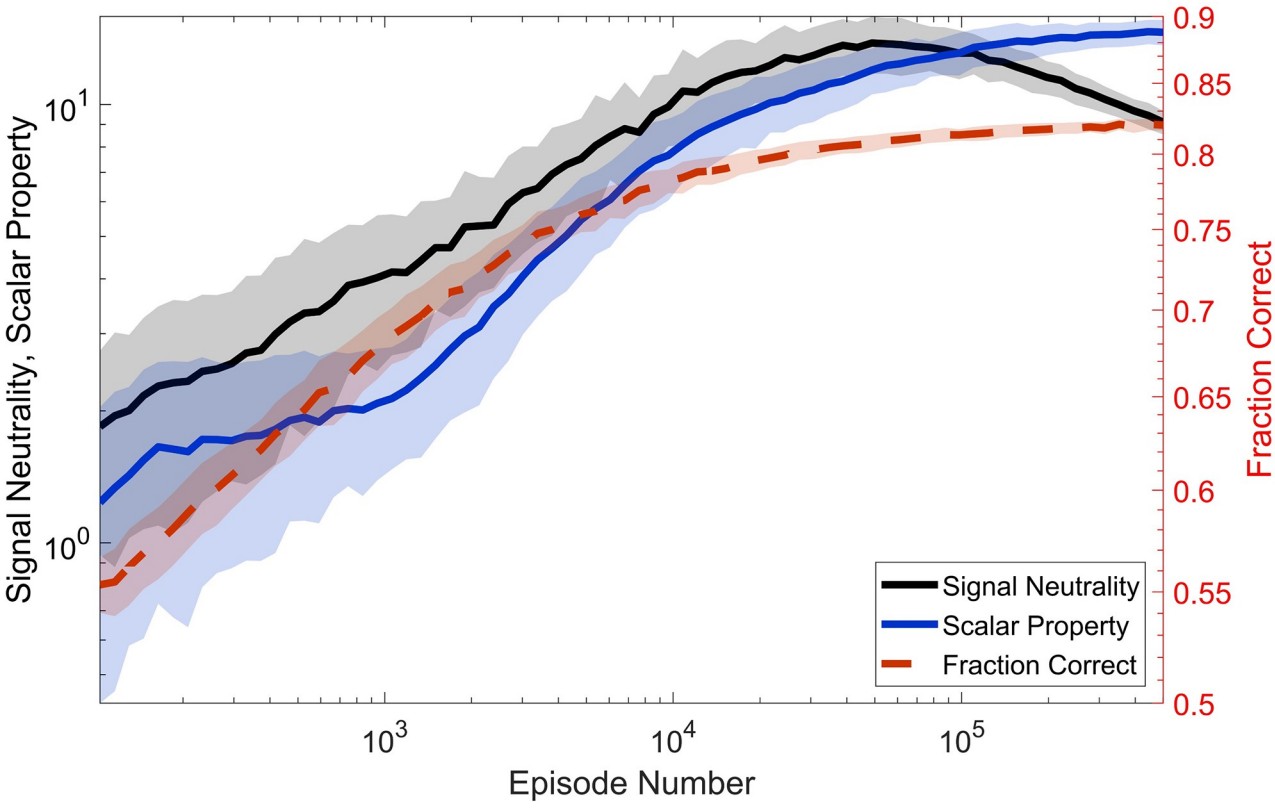

**Fig 7. Signal neutrality and scalar property during training.** Evolution of signal neutrality (black line), scalar property (blue line), and accuracy (dashed red line, scale on the right) as the training progresses. signal neutrality attains a broad maximum where the performance has almost plateaued. Thus signal neutrality can be interpreted as the signature of a 'satisficing' strategy, rather than of an optimal one. The scalar property, on the other hand, keeps growing even for very long training. Yet, the evolution of signal neutrality and the scalar property are highly correlated, suggesting a common origin for the two (see text for discussion).

accuracy (dashed red line, scale on the right y-axis) during training. All the lines are computed by averaging the results of 100 different realisations of the training.

The evolution of signal neutrality and the scalar property are highly correlated for much of the training phase, with an initial fast increase that continues up to about $10^4 - 10^5$ episodes, where the accuracy has almost plateaued (the region used for the results of Figs 4A and 6; note how, after the first $10^5$ episodes, the following $9 \cdot 10^5$ lead to a modest performance gain of $\simeq 1\%$). Such correlated progress naturally hints to a common origin for the two measures, and makes us advance the hypothesis that a behavioural policy displaying these two properties could represent an 'optimal' information-extraction strategy for dealing with a decision task in a volatile environment. It wouldn't be by chance that the agent robustly finds such a strategy by tuning its parameters in a ecologically plausible way.

Yet, after about $10^5$ training episodes, and therefore probably far beyond the experimental training duration, the behaviour of the two curves in Fig 7 starts to diverge. Whilst the scalar property keeps improving, signal neutrality attains a broad peak, after which it gradually breaks down in the face of very modest performance gains. Therefore, the scalar property seems to be more fundamental than signal neutrality, at least for what concerns the strategy asymptotically discovered by the learning agent.

In this sense, signal neutrality cannot be viewed *per se* as signature of an optimal strategy for the agent, but rather of a 'satisficing' one [46]. Faced with a wide distribution of coherences,

the agent pretty quickly finds a robust strategy that, at around decision time, disregards coherence by relying on fluctuations to make decisions, and still ensures a very good performance. Nevertheless, the agent can do slightly better, given enough training time, by giving more weight to the 'drift' component and less to the 'diffusion' component: this is what happens on the far right of the plot. In this region, we postulate, the learning enters an 'overfitting' phase, meaning that the agents become finely attuned to the exact statistics of the task: any slight changes, for example, in the shape of $p(\mu)$ would require many training episodes to revert to a good performance. In this sense, the signal neutral strategy generalises better to novel situations. This is something we plan to study elsewhere. Finally, it is tempting to hypothesise that animal subjects, during perceptual decision experiments, display signal neutrality as a reflex of adopting such a satisficing strategy, given also the high number of training episodes the model needs to refine its strategy beyond signal neutrality.

### 3.4 Collapsing boundaries

It is known that in the presence of a distribution of signal-to-noise ratios and limited decision time, as in the task at hand, the drift-diffusion model is not optimal anymore [15]. More specifically, one ingredient that allows to re-establish optimality is a time-varying threshold. As it has been observed in [9] [11], the optimal decision threshold is not constant when the agent has a finite amount of time to make decisions, but is characterised by a non-monotonic trend across time. This optimal moving threshold is defined as collapsing boundaries. In this sense, the hypothesised optimality of the agent's strategy finds indirect support in the behaviour displayed by the component of $\Delta\Sigma$ that depends only on the passage of time and not on the signal. As we will show, this perception of the passage of time, defined in the model as integration of a constant input over multiple timescales, permits the agent to discover the collapsing boundaries. We rewrite Eq 21 as (see Eqs 10–16):

$$\Delta\Sigma_{\text{right}} = \Delta\Sigma_c^s - \Delta\Sigma^c \tag{23}$$

where:

$$\Delta\Sigma_{\text{right}}^s \equiv \Sigma_{\text{right}}^s - \Sigma_{\text{wait}}^s \tag{24}$$

is a term that provides information on the signal only. And:

$$\Delta\Sigma^c \equiv \Sigma_{\text{wait}}^c - \Sigma_{\text{right}}^c \tag{25}$$

carries information on the passage of time only. We note that on the r.h.s. of Eq 25 we could insert $\Sigma_{\text{left}}^c$ in place of $\Sigma_{\text{right}}^c$ with no notable numerical difference in the result. This is because the right and left choices are *a priori* equivalent in the present task, and therefore the inferred $\theta_{\text{right},\tau}^c$ and $\theta_{\text{left},\tau}^c$ are in fact very similar. For this reason $\Delta\Sigma^c$ does not carry a 'right' label.

$\Delta\Sigma^c(t)$ measures the propensity of the agent at time $t$ to wait for another input instead of making a (either right or left) decision, independently from the signal. Looking back at Eq 23, $\Delta\Sigma^c$ effectively acts as a time-dependent bias term that, in the context of a drift-diffusion model, could be easily interpreted as a time-dependent threshold. Despite the lack of an explicit threshold mechanism for the proposed agent, it is reasonable to expect that the range of values attained by $\Sigma_{\text{right}}^s$ at decision time shifts in accordance with the time-dependent bias. This is indeed the case.

Fig 8A shows (black thick line) the evolution of $\Delta\Sigma^c(t)$ from 0 to $T_{\max} = 2$ s for the proposed agent. In addition, three sample trajectories of $\Delta\Sigma^s(t)$ (coloured lines) are shown from $t = 0$ to decision time (marked by the big coloured circles). The shaded grey area marks the region of

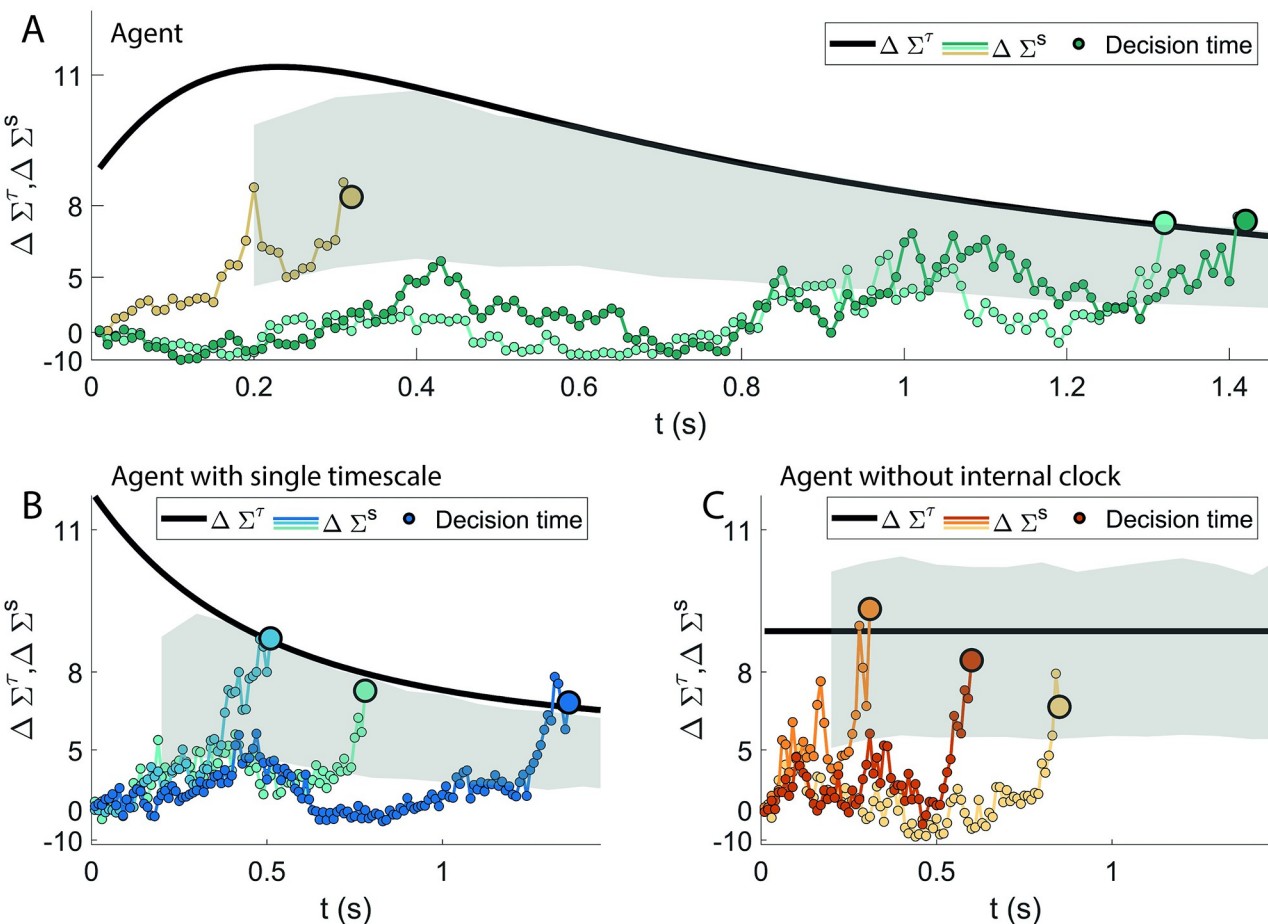

**Fig 8. Collapsing boundaries.** $\Delta\Sigma_{\text{right}}$ (see Eqs 21 and 23) can be decomposed in a signal-dependent part ($\Delta\Sigma_{\text{right}}^s$) and a time-dependent part ($\Delta\Sigma^c$; see Eq 25), that measures the propensity of the agent at each time to wait for another input instead of making a decision. In all panels, $\Delta\Sigma_{\text{right}}^s$ (coloured lines) is depicted for three sample episodes, alongside $\Delta\Sigma^c$ (thick black line). The big coloured circles correspond to the decision times. **A**: The behaviour of the proposed agent. $\Delta\Sigma^c$ acts as a time-dependent threshold: most of the decisions fall inside a strip running parallel to it (the grey area is where 80% of the decisions are made). The resulting boundaries collapse for longer response times. Until about 200 ms, a rise of the effective threshold discourages early decisions. This trend is analogous to the theoretically optimal decision threshold when the trial has a maximum allowed time to make a decision [9, 11]. **B**: The behaviour of an agent with a single integrator (see Section 2.4). Thanks to the internal clock over a single timescale, the agent can implement a suboptimal, monotonically decreasing threshold. **C**: Behaviour of the agent without internal clock, but with multiple signal integrators. The model is unable to exhibit the collapsing boundaries.

values assumed by $\Sigma_{\text{right}}^s$ where 80% of the (correct) decisions are made. As expected, this region mostly run parallel to $\Delta\Sigma^c(t)$, demonstrating how the latter observable can be interpreted as a soft threshold for the decision that arises from the time integrators. Such threshold drops at longer times, a behaviour that finds normative support in the study of perceptual decision making [20, 47]. Conversely, looking at Eq 23, one can view $-\Delta\Sigma^c$ as an 'urgency' signal that pushes for a decision as the episode time elapses, not unlike what has been observed experimentally in the lateral intraparietal area [48].

Fig 8B and 8C report the same analysis for an agent with a single timescale (panel B) and an agent with multiple timescales on the signal but without the internal clock (panel C). It is evident how the agent in Fig 8B exploits the unique timescale available for the internal clock to implement a monotonically decaying threshold. In contrast, the agent without internal clock is unable to create such mechanism, considering that the large majority of the decisions occur in

an area that is parallel to the constant bias $b_{wait} - b_{right}$. The agent of panel C is unable to clearly infer the passage of time from the multiple timescale of the signal. If this limited behaviour can be surprising at first, it can be understood by considering that the present task is highly volatile, with a broad range of signal to noise ratios. Since specific values of the signal integrators $\mathbf{x}_\tau^s$ can be reached rapidly (slowly) for episodes with high (low) coherences of the signal, such features do not constitute a reliable estimate of the passage of time. Indeed, the model in panel C fails in the implementation of any form of urgency signal.

In this respect we want to point out how the soft threshold $\Delta\Sigma^c$ of the proposed model (panel A) does not simply behave as an urgency signal. In fact the decision is made more and more likely as the time passes only after about 200 ms (when $\Delta\Sigma^c$ reaches a peak). Initially, earlier decisions are discouraged by a rise of the threshold. Interestingly, such a non-monotonic trend of the moving threshold has been demonstrated to be theoretically optimal in [9] (see Fig 2B therein; see also [11]).

Even if the models in the references and in the present paper are not structurally equivalent, it is nonetheless striking that the agent can approximate such optimal behaviour by trial-and-error. We note how the monotonically decreasing $\Delta\Sigma^c$ shown in panel B is consequently suboptimal. Thus, the results of Fig 8 demonstrate the necessity of multiple timescales also for an efficient implementation of the collapsing boundaries.

## 3.5 Robustness

The utilisation of a wide range of timescales makes the performance of the agent robust to variation of the task and to the intrinsic noise. This is shown in Fig 9A and 9B. We varied $T_{max}$ (the maximum duration of an episode) and $\sigma_I$ (the standard deviation of the intrinsic noise, $\xi_\tau^s$s and $\xi_\tau^c$s in Eqs 10–15) systematically and, for each value, run the learning process from scratch. The results of the agent are then compared to the models of Section 2.4. While Fig 9 reports the comparison with the single integrators with optimised thresholds, the results for the agent with a single timescale can be found in S1 Text and S1 Fig.

In Fig 9A, as $T_{max}$ increases (and $\sigma_I$ stays at its reference point of 0.02), the fraction of correct responses rises monotonically for all models, with the performance of the agent staying superior on the whole range of $T_{max}$ explored. Two features are noteworthy. First, the lines for the single integrators ($\tau = 0.1$ s and $\tau = 10$ s respectively) cross at intermediate values of $T_{max}$, with the longer $\tau$ surpassing the shorter ones for higher episode durations. Second, the advantage of the proposed agent shrinks in comparison to the longer $\tau$ for longer $T_{max}$. These features have a common origin. From Eq 6, a signal s(t) of mean $\mu$ will asymptotically lead all the integrators to the same (statistically) stationary value of $\mu$, but with different levels of noise. Integrators with longer $\tau$s will have a smaller variance and thus will be more reliable in detecting whether $\mu > 0$ or $\mu < 0$. On the other hand, the time needed to reach the stationary state will be longer for larger $\tau$s. Slower integrators will still be integrating the signal for shorter $T_{max}$ and, as a consequence, their value will carry less information on the $\mu$. Hence, the smaller $\tau$s will dominate for shorter $T_{max}$, the larger $\tau$ for longer $T_{max}$.

In Fig 9B, the level $\sigma_I$ of intrinsic noise is varied, with $T_{max}$ kept constant at 2 s. The performance of the agent (black line) is always substantially higher than that of the single integrators (coloured lines). As expected, performance deteriorates as $\sigma_I$ increases from 0 to 0.2; yet the decrease is only surprisingly slight, considering that the maximum value attained by $\sigma_I$ is comparable with the typical dynamical range of the integrators $x_\tau$. Such range is determined by the distribution $p(\mu)$ (here, a Gaussian of standard deviation $\sigma_\mu = 0.25$). It is then clear that the highest levels of intrinsic noise really affect the typical value of the integrators. This is even more true taking into account that the slowest integrators operate far from the asymptotic

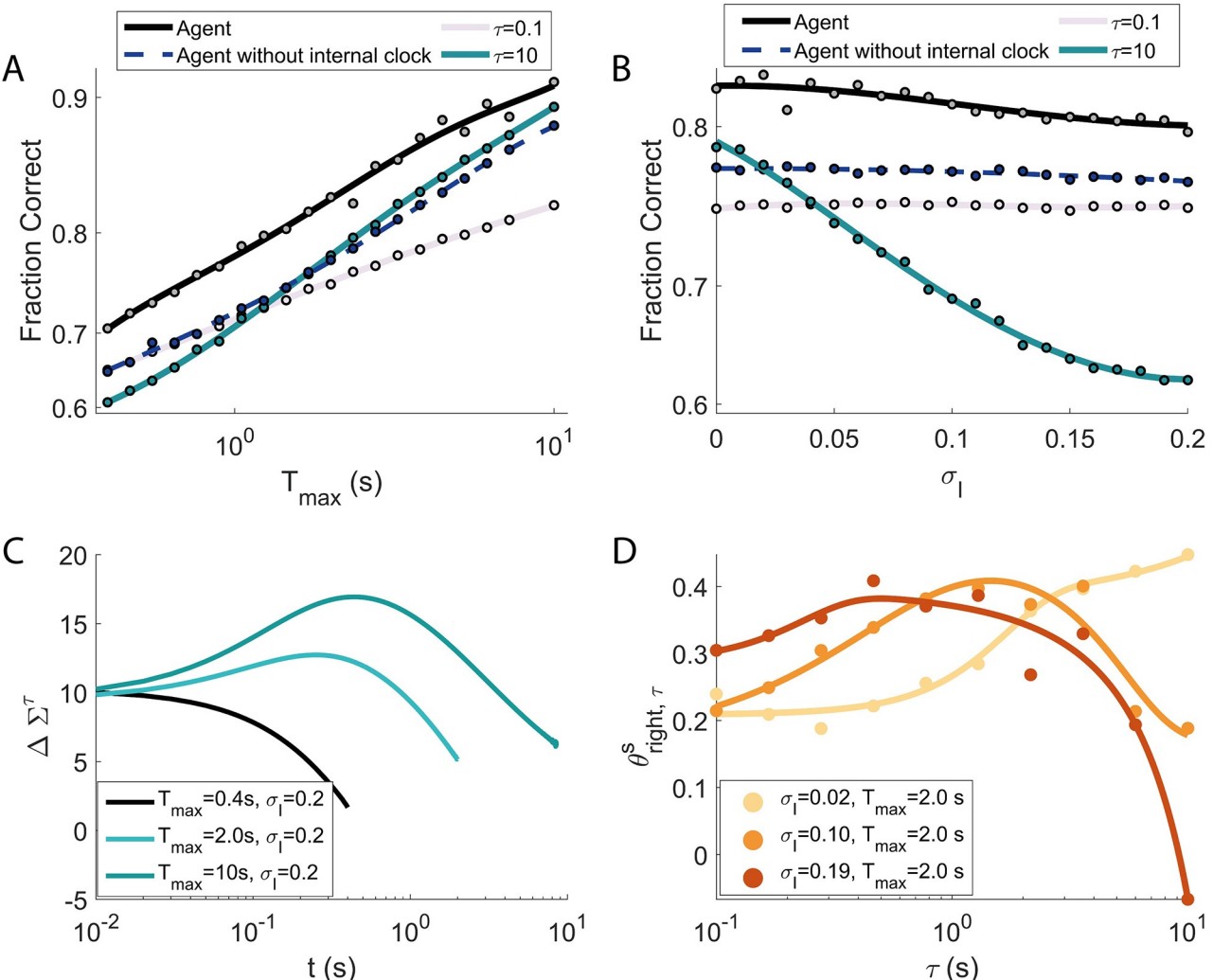

**Fig 9. The wide range of timescales makes the agent's performance robust to variations of the task and to the intrinsic noise. A**: as $T_{max}$ increases, the fraction of correct responses rises monotonically both for the agent (dashed black line) and for all the single integrators, with the performance of the agent staying superior on the whole range of $T_{max}$ explored. **B**: varying the level $\sigma_I$ of intrinsic noise, the performance of the agent (dashed black line) stays always substantially higher than that of the single integrators, notably for stronger noise. As expected, the performance does deteriorate, but the decrease is surprisingly slight, considering that the maximum value attained by $\sigma_I$ is comparable with the typical dynamical range of the integrators $x_\tau$. The performance of the agent without internal clock (dashed blue in panels **A** and **B**) are close (or superior to) the best single integrators reported. However, this agent is more robust than the single integrators over the range of parameters' values considered. Thus, the results show how the model (dashed blue) is able to select the appropriate timescale for different situations (see text for more details). **C**: evolution of the 'moving threshold' $\Delta\Sigma^c$ (Eq 25) for three values of $T_{max}$. For higher values of $T_{max}$ (see also Fig 8), the moving threshold presents a peak whose position shifts with $T_{max}$. **D**: $\theta_{right,\tau}^s$ after training (Eq 10) for different values of intrinsic noise $\sigma_I$ (continuous lines are fourth degree polynomial fits for illustrative purposes). The peak of the lines, corresponding to the most exploited timescale, shifts towards lower $\tau$ values as $\sigma_I$ increases.

value, given the limited integration time. This consideration is clearly reflected in the behaviour of the single integrators. The fast integrators ($\tau = 0.1$ s and $\tau = 2.1$ s) indeed are scarcely affected by the increase in noise. On the other hand, the slowest integrator ($\tau = 10$ s) shows good accuracy for very low levels of noise, but then becomes rapidly ineffective for higher values of $\sigma_I$.

The agent without the internal clock reports robust performance as $T_{max}$ and $\sigma_I$ vary, demonstrating its capability to select the appropriate signal integrator for different conditions.

However, this agent can also report lower performance in comparison to the best single integrators. This phenomenon is a consequence of the reduction of the noise performed on the model with optimised threshold. Indeed, integration of the signal over a single timescale can carry the large majority of the relevant information for a specific simulation. In a specific parameters' setting and in terms of accuracy only, the performance of this agent can be consequently inferior to the model with a single integrator with an optimal value of $\tau$. However, it is clear how the agent depicted in blue is more robust over the range of values shown in the figure, having the possibility to choose the appropriate integration time. For the sake of accuracy optimisation, the role of the reservoir of integrators is to select the appropriate timescale for the considered situation. Instead, the advantage of an internal clock parameterised by multiple timescales is to implement the optimal shape of collapsing boundaries of Fig 8. The latter statement is emphasised by the improved performance of the proposed model also over the agent with a single timescale (S3 Fig), which has information about the passage of time limited over a single $\tau$.

Fig 9C shows the evolution of the 'moving threshold' $\Delta\Sigma^c$ (Eq 25, proposed agent) for three values of $T_{\max}$. For very low $T_{\max}$ (black line) the threshold only decays, always pushing for a decision. For higher values of $T_{\max}$, instead, as we have already seen in Fig 8, the moving threshold initially rises; it reaches a peak and then decays afterwards, making a decision ever more likely. Such peak shifts with $T_{\max}$ and so does, even more clearly, the time at which the threshold reaches back its initial value (around 1 s for $T_{\max} = 2.0$ s, and around 5 seconds for $T_{\max} = 10$ s).

Fig 9D shows $\theta^s_{\text{right},\tau}$ after training ($\theta^s_{\text{right},\tau} \simeq -\theta^s_{\text{left},\tau}$ for the symmetry of the problem after optimisation, as it is shown in Fig 10D) for different values of intrinsic noise $\sigma_I$ (continuous lines are fourth degree polynomial fits for illustrative purposes). Coherently with what we have seen in Fig 9B, the peak of the lines, corresponding to the most exploited timescale, shifts towards lower $\tau$ values as $\sigma_I$ increases.

## 3.6 Evolution during training

Fig 10 illustrates how the behaviour of the agent evolves as it encounters new episodes during learning. Fig 10A shows the performance attained on average for four different values of signal coherence at different times during the training phase. The performance is of course always higher for higher values of coherence ('easier' episodes), and tends to increase monotonically for all the values of coherence during training.

This monotonic trend is not preserved, instead, looking at the average response time (Fig 10B). The response time drops at the beginning of training with values that are very close for every value of coherence. The reason for such behaviour is related to how the agent is initialised. At the beginning, the agent is ignorant about the rules of the task and pre-programmed to make a random choice after having waited for a finite random length of time. Without such random initialisation, the learning would not proceed, since the agent needs to perform actions to learn the relative consequences. While the agent is unable to tell apart signals with different coherences, the response time then decreases. In fact, longer average response times are detrimental due to late responses (no decision before the maximum time allowed $T_{\max}$) that are not rewarded.

This is made clear in Fig 10B, that shows how the fraction of late responses quickly drops to almost zero, and it stays there. Afterwards, the model starts to statistically differentiate between signals with different coherences (the four lines diverge in Fig 10B) and the response time begins to rise. In this regime, waiting means accumulating more information and helps to improve the performance.

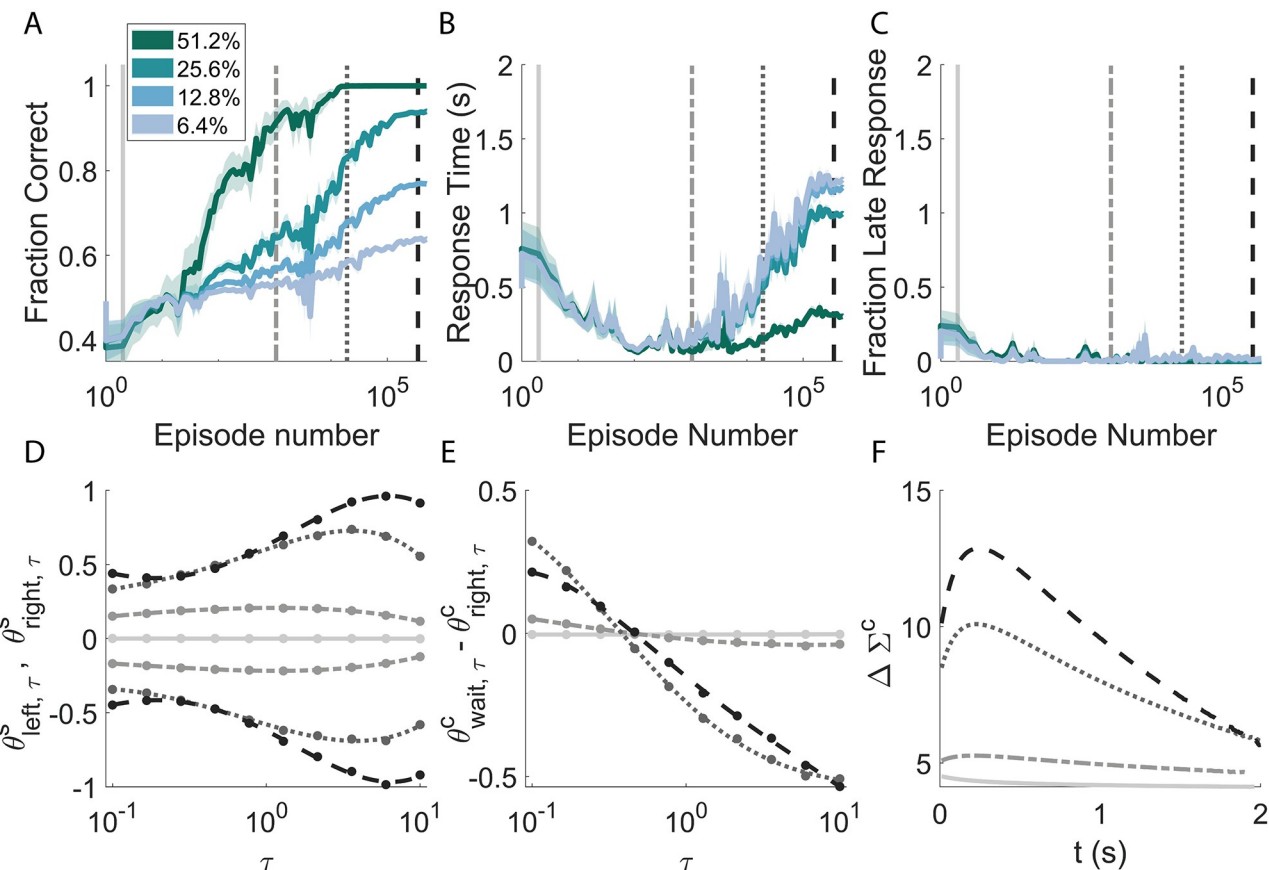

**Fig 10. Learning is characterised by a non-monotonic adaptation of the average response time that is consequent to the necessity of finding a fine balance between integrating information and the cost of waiting to make decisions. A**: Accuracy of the model for signals with different coherences across learning. **B**: Average response times. Trials with increasing level of coherences correspond to greater response times and greater probabilities of 'late' responses. The initial descending trend (around 100 episodes) of the response times common to all coherences is due to the initial ignorance of the agent about the nature of the task, on the tendency to avoid late decisions and to prefer immediate rewards. C: Probability of not making a decision before the end of the episode, i.e. after $T_{\max}$. **D-E-F**: Evolution of different parameters and of the collapsing boundaries for the training instances corresponding to the vertical grey lines of the top panels. **D**: Evolution of the weights corresponding to the integrators of the signal. The weights are positive (negative) for the 'right' ('left') action. **E-F**: Values of the parameters (**E**) defining the contribution of the internal clock to the decision making process and relative collapsing boundaries (**F**).

Fig 10D and 10E show the evolution of $\theta^{s}_{\text{right},\tau}$, $\theta^{s}_{\text{left},\tau}$, and of $\theta^{c}_{\text{wait},\tau} - \theta^{c}_{\text{right},\tau}$ respectively (the shape of $\theta^{c}_{\text{wait},\tau} - \theta^{c}_{\text{left},\tau}$ is similar). The different colours (grey scale) and line styles correspond to the training instances highlighted with the vertical lines of the above panels. Each set of weights is separately rescaled, for each instance, by its absolute maximum value. This has been done to emphasise the relative importance among the parameters rather than their magnitude. At the beginning of training (light grey lines) and despite the rescaling, the weights are close to zero because we initialised the biases of the model at higher absolute values. We chose this initialisation so that the model could exhibit reasonable starting response times without weighting the contribution of the different timescales *a priori*.

As the simulation progresses, the weights for the signal integration toward the 'right' and 'left' actions become stronger while maintaining an approximately symmetric trend with respect to zero (panel D). Also the parameters reflecting the internal clock in panel E grow across training, but fast (slow) timescales become positively (negatively) weighted. Thus, the

weights at a given instance compose an overall descending trend. To understand this, we need to make two simple observations. First, time integrators receive a constant positive signal of one as input. Thus, the sign of the contribution of a time integrator to the decision process corresponds to the sign of its relative weight. Second, such weights are the ones responsible for the implementation of the collapsing boundaries (Fig 10F shows the boundaries corresponding to the considered training instances). By giving positive importance to the fast integrators, which are dominant at the beginning of an episode, the agent is implementing the initial rise of the effective, moving threshold (panel F). In contrast, the negative contribution from the slow $\tau$s is responsible for the collapsing trends reported in panel F.

## 4 Discussion

Decision making and reinforcement learning are fields with overlapping contributions that attempt to demystify how humans and animals make decisions. Our work unifies the two approaches by using a reinforcement learning agent to solve a task reminiscent of a classical perceptual decision making setup, similar to [10, 49–51].

The reinforcement learning agent receives sensory information and information from various "clocks" integrated on multiple timescales. Timescales have been implicit in the reinforcement learning framework, in the context of propagating information about the success (or failure) of the task in cases where the reward is not immediate, see eligibility traces [43, 52]. However, this is not the same as the concept of timescales in this model, where the emphasis is on acquiring and retaining sensory information from the environment, not unlike what happens in the field of Reservoir Computing [53]. We argue that reward maximisation, multiple time constants and perception of time are the fundamental ingredients for faithfully reproducing (i) an optimal decision-making boundary, (ii) the scalar property, and (iii) signal neutrality.

Indeed, the agent learns to solve the task in a relatively small number of episodes, performing better than any single-timescale drift-diffusion integrator while fitting well the psychophysical data. The agent's policy is markedly different from the drift-diffusion model, where a decision happens when one of the integrating processes reach the decision threshold. The reinforcement learning agent makes decisions within short 'active' time windows when fleeting bursts in the probability of choosing an action make that action possible. These "bursts" result from the broad agreement on the decision of many integrators with different timescales, akin to the concept of majority voting. The behaviour of our agent is compatible with the analysis performed in [54] on single-neuron single-trial spike trains in LIP area to uncover sudden activity jumps and their informativeness about choice.

The multiple clock time constants lead to a decision boundary with a shape similar to the theoretical optimal for decisions with bounded time. We demonstrate in simulations that such a complex boundary is not learnable with a single timescale in the clock or without using a clock. The initial increment and then collapse of the decision boundary happen due to the interplay of clock integrators with multiple time constants.

Another direct consequence of the clock with multiple timescales is the scalar property [42, 44], i.e. the ratio of the standard deviation over the mean of the response times remains constant. The removal of the clock results in a fixed boundary over time and cannot exhibit the scalar property. Even a single timescale clock, which leads to a non-optimal decaying decision boundary, fails to capture the scalar property. Our results suggest that an optimal decision-making boundary may lead to the scalar property.

As a side note, and contrary to [42, 44], the experimental results reported in [55] seem to support a linear relationship between response means and standard deviations ($y = ax + b$)

rather than an exact scalar property ($y = ax$). Yet the very low coefficients of variation for fast responses could result from ignoring the effect of non-decision times, which can be assumed to have low variability [20]. In principle, the drift-diffusion model is capable of exhibiting a linear relationship between the mean response time and its standard deviation [55]. Yet, it seems difficult to display the scalar property (see Eq 22).

The multiple timescale signal integrators offer a way to "learn" the integration time constant from the data instead of treating it as a free parameter. Their presence makes the agent robust, as it can perform well with various task difficulties (expressed as a signal to noise ratio or signal coherence). There are optimal integration time constants for specific task difficulties. Varying much the signal to noise ratio would inevitably reduce the performance of an integrator with a single time constant.

A consequence of signal integration with multiple time constants is signal neutrality, the stereotypical collapse of the decision-making signal just before the agent decides. This characteristic is noticeable in the activity of neurons in LIP area during a motion-discrimination task [5, 41]. We can intuitively understand this characteristic in the following way. The agent has to find a policy that works across various coherences. A strategy independent of the specific coherences, if achievable, is an appropriate solution to the problem. The simulations suggest the agent discovers such a policy. To some extent, also the single integrator agents find such a policy. However, if we vary the coherence much, signal neutrality progressively fails.

Fluctuations play a significant role in signal neutrality. And indeed, as far as we can discern, the observation of the phenomenon in [17], where no multiple timescales are present, is rooted in the presence of large fluctuations in the activity traces being averaged. These fluctuations are smoothed out by a first-order filter and a random post-decision time. Nonetheless, they contribute significantly to the observed collapse, as testified by peak values well above the decision threshold.

In summary, our agent learns solely by maximising its reward. There is no strategy *a priori* prescribed, similar to a biological agent during a perceptual decision-making experiment. And yet, our model provides little information about the corresponding mechanisms at the circuit level. Nevertheless, it offers insights into complex processes. We argue that it is a good trade-off between complexity and simplicity [56]: the agent learns when to take actions in an "optimal" way. The learning process suggests that the optimal decision boundary is a consequence of time perception in multiple timescales.

We underline how the proposed agent could be extended to tackle different perceptual decision tasks. For example, being probabilistic, the agent inherently computes an ongoing estimate of the confidence related to each of the possible options. Thus the agent could be presented with the possibility to opt-out from a trial when the choice appears too uncertain. Confidence has moreover been related, in the perceptual decision making literature, to optimal learning [47, 57]. It is interesting to note, in this respect, that the learning rate for the proposed agent is indeed strongly modulated by confidence: an easy correct decision would trigger little learning; on the other hand, a confident but wrong decision would engender large changes in the model's parameters. In [47], moreover, parameters' fluctuations due to the ongoing learning have been shown to account for differences in psychometric curves in an identification task, not unlike the one examined here, *versus* a categorisation task, to which the multi-$\tau$ agent could be adapted with minor modifications. Since we have largely focused the attention on the post-learning phase, the role of such fluctuations in the agent remains an open, and stimulating, issue.

The building blocks of the present model, *i.e.* the signal accumulators, may have biological counterparts [58, 59]. It is possible to use pools of noisy attractors to implement integrators with wildly different timescales, as required by a multi-scale system. Attractor dynamics has

been long one of the main staples of theoretical neuroscience [60]. Several winner-take-all spiking networks capable of implementing a probabilistic classification of the noisy signal have been described in the literature [61, 62]. Therefore we see no conceptual barriers to a more detailed, spiking model mimicking the workings of the agent.

Beyond specific interpretations in this work, we would like to advocate the consideration of multiple timescales in models handling non-stationary and noisy information. There is increasing evidence that performance improves or becomes more robust to changes in the environment when various elements are performing nearly the same task. Adapting to different conditions becomes possible by selectively choosing among those. We notice this general strategy, known as "degeneracy", is present in many biological systems [63–66]. Degeneracy permits rapid adaptation to novel conditions leading to robust performance, adaptability and survivability.

## Supporting information

**S1 Text. Supplementary information containing three sections.** In the first, the actor-critic learning model is described. In the second, we analyse the interpretation of the strategy learned by the agent as a majority vote. In the last, we show how the model maintains high performance despite changes in the distribution of timescales.
(PDF)

**S1 Fig. The agent waits for an alignment of the different integrators before making a decision.** The measures are computed for the episodes where the agent correctly selects the 'right' action. A: Fraction of integrators that are positively contributing to the 'right' action. The measure is aligned with the decision time (extreme right at zero). When a decision is made, more than nine (out of ten) integrators have a positive contribution to the decision on average. B: Probability of the 'right' action as the fraction of positively contributing integrators changes. The probability of making a decision is considerably different than zero when the majority of the integrators align.
(TIF)

**S2 Fig. Surface of accuracy as the number of integrators (x-axis) and maximum timescale (y-axis) vary.** For this specific result, the intrinsic noise has not been rescaled for the different models.
(TIF)

**S3 Fig. Robustness of the proposed model to different parameters' settings and comparison with an agent that exploits a different distribution of characteristic times and an agent with a single integrator (see Section 2.4, Main Text).** A-B: The model with a linear distribution of timescales (red, dashed line) reports comparable performance to the one proposed (black, exponential distribution). This demonstrates that the performance of the proposed agent is robust with respect to changes in the distribution of timescales, assuming that the chosen distribution has time constants over different orders of magnitudes and that is enough dense to cover the range considered. The performance of the agents with single integrators shows similar trends to the one reported in Fig 9 in the Main Text for the integrators with optimised thresholds. Thus, we refer to Fig 9 in the Main Text (Panels A and B) for more detail.
(TIF)

## Author Contributions

**Conceptualization:** Luca Manneschi, Guido Gigante, Eleni Vasilaki, Paolo Del Giudice.

**Data curation:** Luca Manneschi.

**Formal analysis:** Luca Manneschi, Guido Gigante, Paolo Del Giudice.

**Funding acquisition:** Guido Gigante, Eleni Vasilaki, Paolo Del Giudice.

**Investigation:** Luca Manneschi, Guido Gigante, Eleni Vasilaki, Paolo Del Giudice.

**Methodology:** Luca Manneschi, Guido Gigante, Paolo Del Giudice.

**Software:** Luca Manneschi, Guido Gigante.

**Supervision:** Guido Gigante, Eleni Vasilaki, Paolo Del Giudice.

**Validation:** Luca Manneschi, Guido Gigante.

**Visualization:** Luca Manneschi, Guido Gigante.

**Writing – original draft:** Luca Manneschi, Guido Gigante, Paolo Del Giudice.

**Writing – review & editing:** Luca Manneschi, Guido Gigante, Eleni Vasilaki, Paolo Del Giudice.

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
