## [Decision Letter · Decision Letter 0]

7 Oct 2021

Dear Dr Manneschi

Thank you very much for submitting your manuscript "Signal neutrality, scalar property, and collapsing boundaries as consequences of a learned multi-time scale strategy" for consideration at PLOS Computational Biology.

As with all papers reviewed by the journal, your manuscript was reviewed by members of the editorial board and by several independent reviewers. In light of the reviews (below this email), we would like to invite the resubmission of a significantly-revised version that takes into account the reviewers' comments.

As you can see the Reviewers are quite positive about your manuscript. We will nonetheless ask you to address in full all their concerns and suggestions, which we believe will be instrumental in significantly improving the clarity and the scope and the impact of the paper. Please note one of the reviews has been submitted as an attachment. 

We cannot make any decision about publication until we have seen the revised manuscript and your response to the reviewers' comments. Your revised manuscript is also likely to be sent to reviewers for further evaluation.

Sincerely,

Stefano Palminteri

Associate Editor

PLOS Computational Biology

Samuel Gershman

Deputy Editor

PLOS Computational Biology

Reviewer's Responses to Questions

**Comments to the Authors:**

Reviewer #1: This study proposes an interesting extension of standard accumulator models (e.g. drift-diffusion models) to account in a unified framework for three behavioural signatures of decision making: signal neutrality, scalar property and collapsing boundaries. Through an agent that integrates evidence in multiple accumulators running in parallel, each one with different timescales, an alternative strategy to solve the decision problem emerges. Crucially, the authors train the agent using reinforcement learning methods to identify the importance each timescale will have in the overall integration of evidence in a simulated random dot motion experiment. The model seems successful in capturing the behaviour qualitatively (using as main benchmark previous studies, e.g. neuronal firing in LIP) and hints at some testable predictions.

The study is quite comprehensive in the description of the model and characterising its features through varied simulations. I reckon the idea in this work is novel may be of interest for a wide community working on decision making, especially those interested in computational aspects. I suggest some extra points to clarify some aspects of the model and improve the manuscript.

1. The main alternative model to compare the proposed agent is the single-time-scale accumulator. I understand you also used the agent with multi-accumulators, but with a single time-scale, i.e. 10 accumulators with the same τ ((as in Figure 4C?). Is this correct? This alternative model was not as highlighted as the single accumulators, is there a reason for that? Your analytical presentation of the basis of signal invariance and scalar property in the model touches on the necessity of multi timescales, however showing simulations of multiple integrators with identical τ, for some of the other analyses you performed, may be helpful to support the point. This is to dispel the idea that a simple “smoothing” of the signal by averaging multiple accumulators (and not the multi-time scales) could be enough to generate the specific behavioural signatures.

2. Models as the aDDM (Krajbich, Armel, Rangel, NatNeuroscience 2010) or more recently (Jang, Sharma and Drugowitsch, eLife, 2021; or Callaway, Rangel, Griffiths, PlosCompBiol, 2021) have proposed that dynamic attentional fluctuations can affect the evidence integration process, modifying the drift rates or modulating the reliability of the evidence. Based on the premise that it is possible that parameters of the accumulation process could be modulated within a trial, can you comment on the alternative that an agent with a single accumulator but with dynamical time scales could also account for the behaviour captured by multiple-but-static timescales agent (i.e. your agent)? (static understood as an agent whose learning has finished)

3. One of the fundamental aspects of the model is the number of accumulators. In your case, you mentioned the agent comprises 10 integrators. It would be interesting to see in the robustness section of the results how some of the signature behaviours are affected by the number of integrators the agent has. Is there a minimum number of integrators required to generate the behaviour?

4. At the beginning of the paper you hint that this model could help with the issue of a “tuning” phase in which the agents will adjust different accumulators weights for a specific task in a context-dependent way. In this way, the same system could flexibly respond to various task demands. This is a very interesting point, especially considering that standard perceptual experiments do not deal with this stage, all those trials are usually discarded as “training” trials to familiarise the participants with the tasks. It would interesting if you can further elaborate on this point in the discussion. Since it is evident that adaptation in human observers has a completely different scale (not requiring millions of episodes as presented in the RL training), how representative could be this model to characterise that adaptation process?

5. You presented details on the training of the agent to learn the weights, however, it is still not clear to me how it was implemented (e.g. you mention “a large number of episodes” but how many of them?). Was the agent trained on episodes with multiple coherence levels, with an identical number of episodes per level? Were all the different results presented for agents trained on the same episodes? In line 175 you mention that after the training, the same weights are used for the presented analysis. What were the criteria to define the end of the training??

6. About the weights, I did not find a specification of the range they are constrained to (e.g. 0 <w1>

7. The semi-analytical demonstration of the scalar property (line 342) assumes a threshold θ, however in the agent definition is emphasized that the proposed model does not rely on threshold per se. How do you connect the validity of that demonstration with the agent?

8. Abstract and introduction can be restructured to be clearer on the problem the model is trying to tackle. For example, in the abstract, you jump from a very general first sentence on perceptual decision evidence to a very specific second sentence on the agent with multiple timescale integrators, without an intermediate justification. In the introduction, a similar situation is found, with the definition of the features of the agent (line 32) before being clear why that agent is needed at all. This can be confusing the first time one reads the paper. Please try to revise those sections.

9. It may be helpful to dedicate a specific section to the semi-analytical demonstration, instead of it being part of “scalar property” section.

Extra comments:

1. Typo in abstract “The agent discovers a strategy markedly different from the literature “standard”, according to which a decision IS made when the accumulated evidence hits a predetermined threshold”. (Missing 'IS')

2. Line 41: It says “know”, should it be “known”

3. Line 275: “We will focus our attention on the evolution of a key observable in the model”. Is a word missing in this line?

4. Line 500: you mention: “ The intermediate τ = 2.1 s, on the other hand, shows a steadier, intermediate, trend.”, however in figure 9A you show τ = 0.8. Is that correct?

5. Equation 16: I think the symbol to the right side (ς) is not defined in words (although it is understood it is the std.dev.)

6. In equation 25: I guess τ1 and τ2 represent pairs of accumulators with multiple time scales? Additionally, I'm not clear if in the right side of the equation the expression ([1 – exp(-(1/τ1 + 1/τ2)t ]) continues from above or if it is equivalent to the right side of the expression above. Please clarify.

7. Figure 4C: I think a parenthesis is missing in the description of the figure.

8. Figure 4D: this figure is a bit confusing since the numbers increase from top to bottom for the Fraction Correct. Is there a particular reason for that? Also, the lines indicating the agent performance get lost easily with the margins of the figure. It may be a good idea to separate these figures for better visualization.

9. Figure 8: given that you mention that the dot marks the decision time, it is hard to tell if you refer to the big highlighted dot or all the dots that seem to describe the trajectory of the accumulator. Please clarify.

10. Figure 9D: please make bigger the coloured circles in the legends, it is hard to tell which is which.

11. In figure 10, the colour of the legend is very light in comparison to the lines in the figure. It may be a good idea to use the darker version of the colour to ease the identification.</w1>

Reviewer #2: Overall evaluation

The authors attempt to consider an algorithmic (semi-mechanistic) reinforcement learning (RL) model to explain behavior and neural activity observed in a wide class of experiments that require estimating the underlying state of a noisy sensory signal. These experiments have been widely modeled using descriptive mechanistic approaches—specifically the class of accumulator models. The primary contributions here are developing an algorithm that is implicitly close to those used to solve partially observable Markov decision problems, though the explicit relationships are not sufficiently discussed. Further, the authors draw on evidence from behavior and neural data that multiple time scales are used in the brain and that using a reservoir of integrators with multiple time constants they can capture a number of behavioral and neural features observed in experiments. The contributions include a novel RL model, connecting multiple strands in the literature, and connecting the standard integrator models to the normative RL framework. However, the current manuscript does not lay out clearly many of the connections and doesn’t clearly demonstrate the robustness of their approach to the larger class of problems—one of the claims they would like to argue for. That said, I think the core proposal is a significant contribution and with further clarity and some additional characterization would be a valuable addition to the field.

Major Comments

One of the major problems with the paper is that it tries to connect a number of important areas of research without clearly establishing the context and connections. While this doesn’t impact the core proposal embedded in their model, it significantly reduces the impact and clarity of the work. For example, the authors never define or clearly connect to the formal setting for studying uncertainty in RL (and related fields) when the true state of the world is hidden and you only receive noisy observations—the partially observable Markov decision problem. While they mention (and cite) the idea, they are proposing a RL implementation without connecting to this important literature. Another example is that their model is isomorphic to multi-agent RL models, often used to study decision making in economics and computer science. (In this case, each agent would have a different time constant and the arbitration would be via the softmax.) On the biology, they don’t clearly describe the literature on the variation of time constants observed across the brain (e.g. Murray et al., 2014; Bernacchia et al., 2011; Rossi-Pool et al. 2021; etc.) or inference over time for the ‘clock’ processes (so many it’s hard to choose, but in the RL context, Mello et al 2015). If the authors appropriately frame and contextualize their work, it would reduce the sense that this is a somewhat arbitrary model that captures a few behavioral and neural phenomena. Similarly, the relationship between the normative and descriptive could be clarified. There is significant work relating RL models of decision making to the drift-diffusion class of models, including exploring a normative source for collapsing bounds (e.g. Mendonca et al., 2020; Tajima et al., 2016; Stine et al., 2020). Overall, despite the extensive literature cited, the authors need to significantly improve their background description of the problem and the relationship between the biological data, descriptive and normative models.

A second major concern is that the authors don’t provide a clear insight into the decisions that inform the structure of the model. Despite the general insight that using the reservoir of time constants in the brain could provide a general and robust mechanism for solving POMDP problems with opportunity costs (e.g. limited integration time, or fixed length sessions with variable trial lengths, etc.), the model itself is presented without specific motivations for the structure or relation to prior or alternative models. While I don’t feel that the authors need to formally compare their model to alternatives (unless they found it valuable to place it in context), the choices they make feel too arbitrary to the reader and they don’t show how the performance varies under different parameter regimes or reduced versions. This doesn’t provide a sense of the generality or generalizability of their proposed model in the larger context. Specifically, there are a few key questions that should be addressed. Most of these are relatively straightforward to answer or provide the necessary details.

1) Why is it necessary to include a separate time signal? While it is clear the model uses the clock-like integrators to facilitate the speed-accuracy tradeoff, it’s unclear how the model would perform without it in different contexts. Also, it’s unclear why the model cannot extract this time signal from (a transform or function of) the primary integrators which have similar dynamics. Generalized clock-like information is actually relatively unclearly available in biological systems outside of very specific functions (e.g. circadian clock). The kind of temporal inference used in Mello et al., 2015, for example, proposes that general dynamics of sensory and behavioral processing can be used to infer timing and shows the scalar properties required.

2) The relationship between the formal state model, training regime and results. Specifically, often the speed-accuracy trade off is affected by the opportunity to make a rapid choice—even if it’s likely to be incorrect—if the subject can move to a new trial where the difficulty drawn might be easier. Often, this is often controlled for in the primate work cited, but the generality of the algorithm should reflect this as the RL framework is interested in future discounted or average reward rates. Also related to this is a clearer description of the state-transition model used and assumptions about its relationship to time. Classically, RL models have often used tapped-delay lines where time is explicitly in the state space. Here, using eligibility traces, it’s unclear if time is also in the states or not.

3) A clear prediction that often fails for standard drift-diffusion models is the distribution of error times. The authors show a reasonable (compared to data) error trial reaction times, but don’t differentiate this from drift-diffusion models that don’t account for them.

4) How are the parameters chosen and how robust are they? For all the parameters not derived from data or where the values are explained, how did they get fixed and how do they affect the model’s performance. For example, the discount factor, eligibility trace decay, learning rate, etc. Further, if the time constants were chosen to be linear rather than exponentially distributed, how would the model perform? How many time constants are needed in the reservoir? How are the time scales of the perceptual information rate and the reservoir related?

5) The explanation for the consensus periods which lead to the signal neutrality are not sufficiently explained or explored. As the authors point out, other mechanisms can lead to convergence at the time of choice in neural networks and dynamic models. However, the authors want to claim a relationship between the integrators, effective collapsing bound (decision time) and consensus in the weighted integrators. While intriguing, it’s unclear both how reasonable the mapping is between their model’s integrators and the LIP function and activity. Further, would the distribution of errors in time given the ‘windows’ of consensus predict distributions of errors (or choices) uniformly across the trial? If the weights produce different effective balance of short and long time constants at different points in the trial, it seems the distribution of choice and error times might be non-monotonic and different, at least from the description and example plots provided. Finally, more general statistics on the consensus periods across runs would be helpful.

6) A clearer (i.e. likely a figure) to understand the weights distribution through learning and in relation to the integrators at decision time after training would be helpful. The claims about the dynamics require an understanding of the learned weights that are not clearly presented (or how they might vary in the different experiments run).

7) The code available is uncommented, terse and without generation of the plots, and not provided in a reproducible format (e.g. without specific versioning of libraries, etc.). This should be fixed and, ideally, provided in a Jupyter Notebook where readers can manipulate the model easily. Ideally export the corrected GitHub repository and code to a Binder so that it is fully reproducible.

Two additional (but less critical) comparisons would be of significant interest, despite not being strictly necessary to solidify the current work.

1) Extend the model to capture the work on free choice with opt-out (e.g. confidence based) decisions, where the subject has the option to select a lower—but certain—payoff. The POMDP framing and connecting to this line of work on the relationship between these perceptual decisions, confidence and DDMs (e.g. Drugawitsch et al., 2019, etc.).

2) The model’s relationship to analogs of the ‘identification’ and ‘categorization’ task from Mendonca et al., 2020 where the source of noise and effects on reaction time vary depending on the effects of learning related noise vs external sensory limited noise.

Minor comments

The authors would benefit from a simpler and less difficult to read notation. The use of the same variables with similar sub or superscripts vs clearly separable variables makes it difficult to quickly and clearly read the details without lots of checking and extra effort. There are also errors (I believe) in references in places to various variables (e.g. when referring to the different weights in equation 35 and the text in the following sentence seems to be missing a superscript and tilde accent). In general, to facilitate readability by a broader (cognitive or neuroscience) audience, simply making the notation clearer and grouping the details in the methods except where core to the points would be helpful.

What is the formal assumption about the form of the memory (the integrators) in the model with respect to the standard assumptions of RL? Normally, RL solutions are limited to (PO)MDP assumptions where the agent does not explicitly store information in a form that breaks the Markov assumptions (e.g. model-based and model-free approaches commonly used don’t incorporate information from prior states in their policy or value function calculations—except during learning). Here, it’s unclear how exactly the authors are thinking about their integrator processes with respect to the standard (PO)MDP framework.

The relationship between the agent training vs subjects seems unclear. In general, human performance on this class of perceptual decision tasks asymptotes in a short period of time (a few days) where non-human primates require months for similar performance. Presumably, these differences are not related to the actual number of episodes, but rather the ability for humans to understand the structure of the task and goals explicitly, while the primates learn this (via trial-and-error) along with task improvement. A more complete characterization of the learning of the agent and explanation of the behavior in early learning vs asymptotic vs humans and animals might help. Or, simply characterize the agents learning and generalization but don’t relate it to the experimental data—as the mapping is likely complex.

Clarify the framing for optimality (e.g. in the text in lines 468-72) where optimal performance is determined by the objective or cost function assumed. In many of the models, these don’t include aspects like generalization across task variations, etc. The authors seem to want to claim an approximation of some optimal behavior, but it’s not clear if the model is actually specified to be optimal for a more general class of problems or what the assumptions are in the related models and literature they cite and discuss.

What is the relationship to the proposal of leaky (and possibly adjustable) integrators (e.g. Ossmy et al., 2013)?

Reviewer #3: Review is uploaded as an attachment.

**Have the authors made all data and (if applicable) computational code underlying the findings in their manuscript fully available?**

Reviewer #1: Yes

Reviewer #2: Yes

Reviewer #3: Yes

PLOS authors have the option to publish the peer review history of their article (what does this mean?). If published, this will include your full peer review and any attached files.

Reviewer #1: No

Reviewer #2: **Yes: **Eric Edward James Dewitt

Reviewer #3: No
---

## [Decision Letter · Decision Letter 1]

22 Mar 2022

Dear Dr Manneschi, 

Thank you very much for submitting your manuscript "Signal neutrality, scalar property, and collapsing boundaries as consequences of a learned multi-time scale strategy" for consideration at PLOS Computational Biology. As with all papers reviewed by the journal, your manuscript was reviewed by members of the editorial board and by several independent reviewers. The reviewers appreciated the attention to an important topic. Based on the reviews, we are likely to accept this manuscript for publication, providing that you modify the manuscript according to the review recommendations.

Sincerely,

Stefano Palminteri

Associate Editor

PLOS Computational Biology

Samuel Gershman

Deputy Editor

PLOS Computational Biology

[LINK]

Reviewer's Responses to Questions

**Comments to the Authors:**

Reviewer #1: The authors have made good work addressing the points and improving the manuscript. I would also like to highlight the quality of the figures. I am making some final extra suggestions just to clarify some points in the latest version:

1. In line 42 you state that “To avoid the issue of tunning a single parameter to the data…”, however, I think that at that point of the introduction it is not fully clear yet why it is an issue at all (especially for someone outside the field of decision models). It may be a good idea to restructure that paragraph, so the disadvantage of a single parameter is clear. Maybe moving the paragraph on the biological evidence supporting multiple timescales ahead can help.

2. Thank you for including extra information on robustness and the number of integrators (section 5.2). It would be great if you can add a plot similar to Figure 12, not only for fraction correct, but that also shows how signal neutrality and scalar property are affected by those two factors.

Extra comments:

1. In the abstract, in the first line, I’m not sure in which sense the word “underline” is used.

2. In line 8, when it says, “on the other hand”, it is not clear what you are contrasting.

3. In line 37: “The task is reminiscent of the famous Shadlen experiments…” please add the reference.

4. In subtitle 2.2. typo “Relationshipt”

5. In line 292: “The comparison with the fixed-t observer sheds lights”, should be “sheds light”

6. In line 309, it is mentioned that in Figure 3b, the agent is represented by a dashed line, when there is no dashed line in that panel.

7. In line 364, again it is referred to a dashed line in figure 5, when the figure does not show it in that way.

8. Line 484: typo “such a non-monotinic trend…"

9. Line 484: typo “demonsrated ”

10. Line 508: typo “asimptotically”

11. Line 639: typo “scaler property”

12. In the supplemental information: Typo in the first line, page 2: “In cotrast”

13. In the paragraph of the second page of the Supplemental info: “Fig. 12 report the performance of two agents, one with the exponential distribution of τ adopted in the paper (black line as usual)…”. I think the description does not match the figure. Should it be Fig 13 instead?

14. In Figure 2, the caption does not include a description for s(t).

15. In Figure 2, typo in “spceific “

16. In figure 9D: Thanks for making the circles in the legend bigger. However, it may still be difficult to identify the colour of the circles in the plot. Maybe, increasing slightly their size or removing the margins of the dots can help.

17. In figure 10: from just looking at the figure, it may take a while to notice that the grey lines correspond to specific episode numbers. It may help to add the episode number as a legend for the panels D-E-F panels or use more distinct colours for the lines.

Reviewer #2: Overview

The authors consider an algorithmic (semi-mechanistic) reinforcement learning (RL) model to explain behavior and neural activity observed in a wide class of experiments that require estimating the underlying state of a noisy sensory signal. These experiments have been widely modeled using descriptive mechanistic approaches—specifically the class of accumulator models. The primary contributions here are developing an algorithm that is implicitly close to those used to solve partially observable Markov decision problems and that suggest a reservoir of time constants better explain features of the experimental data than other (sometimes more normative) approaches proposed. The contributions include a novel RL model, connecting multiple strands in the literature, and connecting the standard integrator models to the normative RL framework. The revised manuscript addresses the majority of concerns highlighted by the reviewers in the first round. The manuscript still has a few areas where some simple improvements would likely increase the readability and impact, and the code provided should still be commented more substantially. However, these problems should not prevent publication.

Comments

The manuscript is vastly improved and most of my major objections were well addressed and I believe those of the other reviewers. In many places, the text is more complete and the additional model comparisons address a number of the questions open in the original manuscript. I also think the authors did a good job in responding to the reviewers' questions and suggested changes and the additional supplementary information is valuable. I believe it is acceptable for publication, but I have a number of comments that could improve the manuscript and clarify a few remaining questions. One example is that the core operational definitions for the criteria used to evaluate the model and its relationship to the prior literature could be introduced more clearly. The term “signal neutrality” is used both as though it is a common technical term in the manuscript and in a way that is confusing to the reader. The other two criteria are defined in terms that are common in the field and well understood casually, though the use of ‘scalar property’ without mentioning Weber (scaling) and the large related timing literature directly in a manuscript targeted to a wider audience is a major oversight. In general, as was mentioned by Reviewer #3, I think the manuscript still is written in a form that often mixes broad assumptions about the reader's knowledge with overly complex and technical language—but unnecessarily so for the purpose. While a complete solution would likely involve restructuring the manuscript to separate the core ideas and results from the technical material, it would suffice to at least have a set of colleagues who are not familiar with the specifics to note all the places where clarifications or simplifications would benefit the reader. I don’t feel that it is the reviewers jobs to do copy editing, so I will not provide that level of detail, but I will point out a few specific problems I think should be addressed.

First, ‘Shadlen-like’ as a description of the class of experiments is both not technically correct nor clear to a reader unfamiliar with the specifics of those experiments, while the model and general points are widely applicable. The correct citation for the specific random dot kinematogram would be Newsome & Pare (1988) or perhaps Newsome, Britten & Movshon (1989) along with the related papers. However, the general question of integrating sensory information is much larger and most of the results in the paper extend beyond the narrow MT/LIP random dot kinematogram task literature. The same task has been used extensively and similar tasks with different superficial structure show the same basic behavioral signatures e.g. in olfaction, somatosensation, etc. In short, it would be better to describe the class of tasks and mention the specific experiment you replicate rather than using a reference that makes assumptions of the reader's knowledge and implies perhaps a narrower range of applicability.

It would be nice if Figure 1 graphically described the complete model. By this I mean that the choice (multiple-logistic/Boltzman/softmax) function is missing. As is the a graphical description of the environment’s relation to choice and reward feedback (i.e. consider the corresponding actor-critic agent graphic from Sutton and Barto). While the current figure is sufficient, a reader wanting to understand the model clearly needs more.

Figure 2 seems like it might be clearer to the reader in the context of the rest of the paper if the non-exponentiated (i.e. summed weights, rather than probabilities) were used. This would also correspond more to the scale used for the colored circles.

Signal neutrality is an unclear term that is not found in the literature nor is it easy to interpret in the context as a natural phrase in English. The signal is the external property (i.e. the random dots) and not the variable represented by the accumulators or neurons. I believe the authors intend to imply ‘signal strength neutrality’, that is, that the coherence differences are no longer observed in the internal variable at the time of decision—regardless of whether it is approaching an absorbing boundary or if it is caused by the mixing of time constants in their model. I don’t want to propose a specific solution for the authors, but perhaps they could find a term from the literature or at least introduce this term very clearly at the conceptual level. Currently, it’s introduced in a narrow context of neural signals. The same properties are seen in other decision making models (e.g. Wang 2002). Specifically, regardless of the terminology, clarification of the relationship between the way the author’s model implements a decision and the others and the respective models’ relationship to this property should be more clear to the reader. For example, the neurons are thought to be reaching a saturation (perhaps maximal firing rate) prior to motor command initiation which resets in some interpretations. Other models predict this property trivially (in that they define the decision as reaching a common threshold. The criteria and its relationship to models and the data should be clear along with the implications.

It is unclear why the signal neutrality property drops at the end of training and how flat the changes in expected reward are at that point. Is this a possible overfitting or other problem with the reward gradient? Is the model actually still improving while signal neutrality is declining for a reason that can be explained? The current text does not provide much insight and seems to imply that the model has not asymptoted (“very modest performance gains”), but it’s unclear then how to interpret if the signal neutrality property is being appropriately related to the monkeys (asymptotic) performance, and more critically, just what is changing that allows the performance gains at the expense of the signal neutrality. I think there is no particular justification provided for the length of training (or point selected along the relative flat performance period) for analysis. Nor does it seem necessarily pertinent to the story, so perhaps Figure 7 and the accompanying text could be pushed to supplementary information or at least moved to section 3.6 with Figure 10.

Mention Weber law and Weber scaling when introducing and contextualizing the scalar property. I note the word “glaringly” in the abstract when describing this which seems quite odd. But more generally, this is a broad and important literature of observed properties (with varying underlying models) that should and could be at least contextualized with the ‘scalar property’ used here.

The description of collapsing boundary property and Figure 8 is a bit confusing for the reader as presented. In Figure 8, the propensity from the clock follows the expected (near optimal) shape of the collapsing bound, but the actual decisions made by the agent are presented showing the signal evidence, which is closer to a flat distribution of evidence—if I’m following the subscripts correctly and the filled circles only represent the stimulus weights for the choice. But this is what you would expect from the combination, and the description of the clocks providing a ‘soft’ bound effect. However, the text seems to imply that the actual sensory evidence weights are also showing the effect, which seems counterintuitive—as they have been mathematically separated in the variable shown to exclude the effects of time, other than via the distribution of integrator time constants. In the end, they vary slightly across time in the direction you might expect from the optimal POMDP model, or the clocks alone. But if I were to take the mean decision points (inferred from the 80% shaded band), the weighted evidence would barefly change with time compared to the variance of the decision times. It’s unclear from the description and the few examples shown (and the 80% band) if the pure sensory evidence weighted integrators are an additional contribution to the soft collapsing bound property or if it would be better explained (or at least described) as a model that has a collapsing bound propensity via an effectively separate mechanism from the sensory evidence? In which case, it’s not clear what the samples and 80% band is conveying. I also note that the impact of the time on the decision times given the parameters used in the current simulations may be small. I assume that different parameters (e.g. reward function) or modeling a different task variant (e.g. one with an opt-out as seen in the literature) might demonstrate the effect of this propensity more clearly. However, this seems likely just the way the bound is described relative to the model’s decision points in the figure. The comparisons are interesting and the main point seems to hold as long as the clock integrators show the curves presented in the black lines.

In Figure 9D, isn’t there a natural prediction (e.g. from the optimal POMDP or relative to a fixed sumed clock from 9C) that would provide true predictions rather than the polynomial fits? Or, to put it another way, shouldn’t the function of the weights be smooth over the times for for large samples so that you see a clear peak that shifts? This may be just a function of the noise relative to the number of models trained or the amount of data from the model?

Figure 10 needs a matching color scheme or legend for D,E,F.

Overall, the learning results are interesting, however they suggest to me (i.e. F7 and F10) that the model struggles a bit to converge and is perhaps taking a trajectory that is caused by the lack of constraints on the weights. Reviewer #1 also asked about the weights and you responses suggests you didn’t consider the fact that the weights necessarily are only relative—that is, while they can vary across the whole number line in your current definition, by the constraints of the learning algorithm the critic weights are implicitly constrained to rescale the average evidence and clock integrators to the scale of the reward. On the policy side, there is no such implicit constraint as the relative weights enter into the softmax in a form where they can trade off across the (absolute) scale. Given that for the data the agent sees and the noise in the integrators and signal, it might be better for the study of learning to constrain the actor (policy) weights to sum to one (or zero) and always re-normalize them to a unit vector. This would ensure the reward gradient is driving the relative weighting. As the reward function is trivial and the basis set large, perhaps a similar consideration for the critic is appropriate. In any case, I would probably move the learning to supplementary if left as is, despite it being interesting and generally supportive of the overall story.

Check section 5.2, F.igure 12 is referenced when Figure 13 is meant throughout the text. The (actual) Figure 12 would be more useful if it showed 1-21 integrators, as above 21 there isn’t any information and given the task, seeing the degradation to 1 integrator is more useful. Similarly max time scale from sub 1 to 10 would be more informative.

Finally, overall the technical notation and choice of language in describing the model mixed with connections to the literature and experimental data makes the readers job difficult. When you read much of the manuscript and look at the figures, you need to refer to the specific definitions of the objects and the authors choice of symbols, etc. It was already noted in the first reviews that the language and copy editing should be checked and that it was difficult. This seems unnecessary, in that in many places the technical elements—that is the mathematical formalism and choice of notation—are not that critical to any of the author's points. Further, as there is no analytical elements to the paper, and the core models and definitions are common enough in the computational neuroscience literature and the surrounding referenced literature, from this reviewers point of view, much of the technical definitional math could be separated from the descriptions of the computation and algorithms and the conceptual objects and where possible. The use of words rather than symbols for non-canonical elements—at least in the main text—for example. As an example, there are 10 integrators and 10 clocks. Why “x_{tau}(t)” and “x+{tau}^{Tau}(t)”? Why not “integrator_{tau}(t)” and “clock_{tau}(t)”? If the authors want to make their ideas accessible to a wide audience, they can make it easier for them without diminishing the technical clarity for those who wish to evaluate the details. After all, supplementary information could be used and the code is shared—in fact, Jupyter notebooks include latex rendering so the full math should be there too. However, this is as much a personal style comment and a suggestion for trying to increase the audience and impact as a criticism. The current draft is an acceptable presentation of the ideas and results.

Which brings me to the final comment, the Jypter notebook now in the code repository is a vast improvement, but the primary code and the notebook could both provide exactly the level of beneficial clarity requested above by including actual comments and clearly explicating text cells. Why the authors seem intent on providing all the code but making exploring the model difficult is a mystery.

Reviewer #3: Thank you for the revision. Although the manuscript would still benefit from proof-reading (example, line 99), I think the manuscript has been improved much in terms of clarity, together with other reviewer's comments. My comments have been fully accounted for.

**Have the authors made all data and (if applicable) computational code underlying the findings in their manuscript fully available?**

Reviewer #1: Yes

Reviewer #2: Yes

Reviewer #3: Yes

PLOS authors have the option to publish the peer review history of their article (what does this mean?). If published, this will include your full peer review and any attached files.

Reviewer #1: No

Reviewer #2: **Yes: **Eric Edward James DeWitt

Reviewer #3: No

Figure Files:

Data Requirements:

Reproducibility:

References:

---

## [Editor Report · Decision Letter 2]

8 Jun 2022

Dear Dr Manneschi, 

We are pleased to inform you that your manuscript 'Signal neutrality, scalar property, and collapsing boundaries as consequences of a learned multi-time scale strategy' has been provisionally accepted for publication in PLOS Computational Biology.

Best regards,

Stefano Palminteri

Associate Editor

PLOS Computational Biology

Samuel Gershman

Deputy Editor

PLOS Computational Biology

---

## [Editor Report · Acceptance letter]

26 Jul 2022

PCOMPBIOL-D-21-01431R2 

Signal neutrality, scalar property, and collapsing boundaries as consequences of a learned multi-time scale strategy

Dear Dr Manneschi,

I am pleased to inform you that your manuscript has been formally accepted for publication in PLOS Computational Biology. Your manuscript is now with our production department and you will be notified of the publication date in due course.

With kind regards,

Marianna Bach
